# CompactDP: Category-Aware Feature Compactness for Differential Privacy

## Abstract

The rapid growth of AI models raises critical privacy concerns due to their tendency to memorize training data, making them vulnerable to extraction and membership inference attacks (MIAs). Traditional privacy-preserving methods like DP-SGD often degrade model utility, limiting their applicability in sensitive fields. We observe that compact class-wise feature manifold embeddings inherently reduce privacy risks by smoothing probability density functions (PDFs), which diminishes the influence of individual training samples and lowers memorization. Leveraging this insight, we propose *Class-wise Compactness for Privacy* (CompactDP) method, a noise-free feature contraction framework that directly addresses the root cause of privacy leakage, sparse, high-dimensional features, via feature contraction rather than relying on gradient noise. CompactDP can be trained independently and achieves a superior privacy-utility trade-off, with empirical privacy guarantee comparable to DP-SGD ($\epsilon = 1$). CompactDP attains 95.82% accuracy while limiting MIA risk comparable with DP-SGD ($\epsilon = 1$) on CIFAR10. Notably, leveraging the optimized feature embedding, DP-SGD maintains robust model utility while preserving rigorous privacy guarantees across varying privacy budgets. Extensive experiments on FashionMNIST and MedicalMNIST further validate its favorable utility-privacy trade-off across diverse metrics.

## 1 Introduction

The emergence of billion-parameter neural networks has revolutionized machine learning, delivering state-of-the-art performance across diverse domains (Zhai et al., 2022). However, these models exhibit concerning memorization capabilities (Zhang et al., 2021), creating significant privacy risks through extraction attacks (Carlini et al., 2021) and membership inference. This vulnerability is particularly critical in sensitive sectors such as healthcare and finance, where models trained on private data must maintain rigorous privacy guarantees without compromising utility. Differential Privacy (DP) (Dwork et al., 2006) has emerged as the gold standard for privacy-preserving machine learning. Nevertheless, the practical implementation of DP in deep learning, particularly through Differentially Private Stochastic Gradient Descent (DP-SGD) (Abadi et al., 2016), faces several fundamental limitations. Current approaches suffer from three critical shortcomings (Kulynych et al., 2022): (1) Gradient perturbation in large models with high-dimensional features disproportionately degrades utility with accuracy drop at most 5+% under ($\epsilon = 1, \delta = 1e - 5$) settings; (2) Standard DP mechanisms apply uniform protection across all classes, disregarding inherent class-wise privacy leak risks.

Motivated by the privacy-utility dilemma, we establish a fundamental connection between class-wise feature manifold embedding compactness and privacy vulnerability. In Fig. 1, we illustrate the fundamental mechanism of our feature space contraction approach. The transformation process systematically draws sparse samples from low-probability regions toward high-density areas within each class distribution, resulting in significantly more compact class-wise PDFs. As shown in the left subfigure, initially dispersed samples (represented by cool colors) undergo a contraction process that redistributes them toward the dense core regions (warm colors) of their respective class distributions. This geometric transformation reduces the effective diameter of each class cluster while preserving the inherent manifold structure. The right subfigure demonstrates the final contracted state, where each class forms a compact, well-defined manifold with minimal peripheral dispersion.

This contraction mechanism directly enhances privacy protection by reducing the presence of outlier samples that are particularly vulnerable to membership inference attacks. The compactified feature distributions minimize the surface area exposed to potential adversaries while maintaining the discriminative power necessary for accurate classification. The resulting geometric configuration formalizes privacy preservation through intrinsic feature space optimization rather than external noise injection. Our experiments on CIFAR-10 reveal that *privacy risk scales super-linearly with class feature space PDF diameter $d_c$*: classes with sparse manifold distributions (e.g., *Bird*) exhibit higher empirical leakage than compact embedding (e.g., *Automobile*). This class-wise manifold perspective yields a critical insight: strategically contracting feature diameters can simultaneously enhance privacy and utility. Different from cluster based methods (Caron et al., 2018), our framework can keep the low-dimensional class-wise embedding manifold in the contraction learning stage while other methods may disrupt the manifold structure in the clustering learning process. The learned contracted embeddings in low-dimensional manifold can be seen in Fig. 3b.

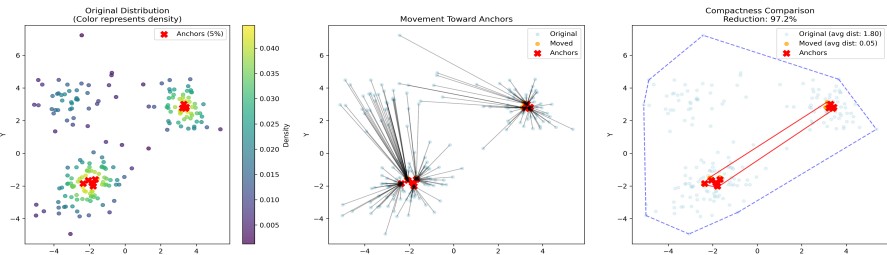

Figure 1: Illustration of feature space contraction through density-based compaction. Low-density peripheral samples are progressively drawn toward high-density core regions, reducing class-wise PDF diameters and minimizing privacy leakage. The left panel shows initial sparse distributions, while the right panel demonstrates the resulting compacted feature clusters with reduced surface area in low dimensional manifold space and enhanced privacy protection.

The class-wise manifold characteristics, including distribution, diameter, and dimensionality, serve as critical variables in privacy protection. By strategically reshaping data distributions through feature contraction, we achieve stronger privacy guarantees without sacrificing model utility, thereby breaking the traditional privacy-utility trade-off. This work introduces a new privacy-utility learning paradigm: shifting privacy protection from passive noise injection to active optimization of manifold feature structures. We present three contributions: first, we propose CompactDP, a manifold embedding feature space contraction framework with rigorous theoretical guarantees that enables privacy amplification via feature contraction; second, we demonstrate feature embedding quality's pivotal role in the privacy-utility trade-off—our framework retains generalizable rare-example information (Feldman, 2020b) while suppressing privacy-sensitive memorization artifacts, offering actionable insights into balancing these objectives; third, we show that optimized feature embedding allows DP-SGD to maintain robust utility and privacy guarantees across varying budgets, with our approach delivering strong trade-off performance across benchmarks and architectures and challenging the conventional strict utility-privacy trade-off assumption in differential privacy.

## 2 RELATED WORKS

The connection between class-wise feature embedding manifold and privacy vulnerability has gained increasing attention. (Sanyal et al., 2022) revealed that sparse class distributions heighten membership inference risks, While Berrada et al. (Berrada et al., 2023) optimize neural architectures to improve DP-SGD's accuracy-fairness balance, CompactDP introduces geometric feature contraction as an orthogonal approach that restructures the representation space itself to achieve stronger privacy-utility trade-offs. Subsequent work exploited this insight through regularization (Farrand et al., 2020) and outlier suppression (Bagdasaryan et al., 2019), but these approaches lacked theoretical grounding. (Hardt & Talwar, 2012) first linked global sensitivity to feature distances but required impractical noise levels. Our work bridges these critical gaps by establishing the first theoretical framework for linking class-wise manifold geometry with privacy utility trade-off, where contraction parameters dynamically adapt to local density, enabling precise mitigation of category-specific

vulnerabilities without utility degradation. Our work challenge the established understanding of the privacy–utility trade-off, which states that improving utility typically requires memorization, particularly of outlier or rare samples, which inherently reduces privacy. (Feldman, 2020a; Feldman & Zhang, 2020).

**Our Key Differentiators** While previous works have established the importance of feature quality for privacy, **CompactDP** introduces the novel paradigm of **actively contracting feature manifolds** as a privacy-enhancing mechanism. Unlike methods that rely on existing representation quality, we systematically optimize the feature geometry to simultaneously reduce gradient sensitivity, minimize outlier memorization, and provide inherent privacy benefits that complement formal DP guarantees. Our approach represents a shift from passive noise addition to active geometric transformation of the feature space. Kulynych et al. (Kulynych et al., 2022) highlight the critical issue of disparate vulnerability in privacy protection, showing that membership inference attacks affect demographic groups unevenly. This underscores the limitations of one-size-fits-all privacy mechanisms. **CompactDP**'s class-adaptive contraction naturally addresses this concern by ensuring consistent privacy protection across all classes through individualized geometric transformations, potentially mitigating disparate vulnerability.

More related works can be found in App. K.

## 3 THEORETICAL FOUNDATIONS

### 3.1 PRELIMINARY

Given a backbone network $f_\theta(\cdot)$ and input samples $\mathbf{x}_i$, we extract features:

$$\mathbf{z}_i = f_\theta(\mathbf{x}_i) \in \mathbb{R}^d \tag{1}$$

where $d$ is the raw feature dimension.

For each class $c$, we select anchor points $\mathcal{A}_c$ as the top $\gamma\%$ of samples with highest PDF to represent the manifold structure of the class-wise embeddings.

$$\mathcal{A}_c = \{\mathbf{z}_j : y_j = c, p_c(\mathbf{z}_j) \geq Q_{1-\gamma}(\{p_c(\mathbf{z}_k)\}_{k:y_k=c})\} \tag{2}$$

where $Q_{1-\gamma}$ denotes the $(1-\gamma)$-quantile.

We define a class-wise feature contraction function $g_\phi(\cdot)$ implemented as a neural network:

$$\hat{\mathbf{z}}_i = g_\phi(\mathbf{z}_i) \tag{3}$$

To achieve intra-class compression, in the implementation we design a class-wise PDF contraction loss function to replace the above compactness term:

$$\mathcal{L}_{compact} = -\sum_{\hat{\mathbf{z}}_i \in D'} \mathbb{1}_{\{y_b=t\}} K_h(\hat{\mathbf{z}}_i - \mathbf{a}) \tag{4}$$

where $\mathbb{1}_{\{y_b=t\}}$ is an indicator function selecting references points from the same class as $\mathbf{a}$ in the contracted feature space $D'$. $t$ is the class to be contracted. $K_h(\cdot)$ is a kernel function with bandwidth $h$. The logarithmic transformation of probability values stabilizes training and prevents numerical underflow during backpropagation. This loss function encourages compact feature clusters within classes. The whole framework is depicted in Fig. 2 and further explained in App. M. Our focus is on training the feature contraction network parameterized by $\phi$, which is parameter-free in the classification layer. We estimate the PDF $\hat{p}(x)$ for each class based on the fine-tuned feature space, e.g., a 768-dimensional space for ViT-B/16. During inference, a test sample x is classified by assigning it to the class with the highest estimated PDF value:

$$\hat{y}(x) = \arg\max_j \hat{p}_j(x), \tag{5}$$

Notably, our framework is noise-free during training and independent of DP-SGD-based approaches. Furthermore, it can be integrated into existing DP-SGD workflows by appending an additional classification layer, where parameters are updated via noisy gradient updates.

## 3.2 PROBLEM SETUP

Consider a dataset $\mathcal{D} = \{\mathbf{x}_i, y_i\}_{i=1}^n$ with samples $\mathbf{x}_i \in \mathbb{R}^d$ and labels $y_i \in \{1, \ldots, C\}$. Let $g_\phi : \mathbb{R}^d \to \mathbb{R}^p$ be a feature contractor. For each class $c$, define:

- Class-wise features: $\mathcal{F}_c = \{g_\phi(\mathbf{z}_i) : y_i = c\}$
- Class diameter: $d_c = \max_{\mathbf{z}_i, \mathbf{z}_j \in \mathcal{F}_c} \|\mathbf{z}_i - \mathbf{z}_j\|_2$
- Class-conditional PDF: $p_c(\mathbf{z}) = \frac{1}{|\mathcal{F}_c|} \sum_{\mathbf{z}_i \in \mathcal{F}_c} K_h(\mathbf{z} - \mathbf{z}_i)$

where $K_h(\cdot) = \frac{1}{h^p} K\left(\frac{\cdot}{h}\right)$ is an $L$-Lipschitz kernel and $h$ is the bandwidth. After *class-wise feature contraction*, we obtain contracted features $\mathcal{F}'_c$ with diameters $d'_c \ll d_c$ and PDFs $p'_c(\mathbf{z})$.

**Definition 1** (Feature Space Vulnerability Measures). *The privacy vulnerability of class $c$ is characterized by:*

$$\text{Diameter Contractor Ratio:} \quad \eta_c = d'_c / d_c \tag{6}$$

$$\text{Local Density:} \quad \rho_c(h) = \mathbb{E}_{\mathbf{z} \sim p_c}[p_c(\mathbf{z}; h)] \tag{7}$$

$$\text{Contraction Factor:} \quad \kappa_c = 1 - \eta_c \tag{8}$$

*Smaller $\eta_c$ indicates stronger contraction and lower vulnerability.*

**Definition 2** (Class-Conditional Kernel Density Mechanism). *Given a dataset $\mathcal{D}$ partitioned by class labels $c \in \mathcal{C}$, the class-conditional kernel density estimator mechanism releases:*

$$\mathcal{M}_c(\mathcal{D}) = \frac{1}{|\mathcal{D}_c|} \sum_{x_i \in \mathcal{D}_c} K_h(\mathbf{z} - g_\phi(x_i)) + \mathcal{N}(0, \sigma^2 I) \tag{9}$$

*where $\mathcal{D}_c = \{x_i \in \mathcal{D} : y_i = c\}$ is the class-specific subset, $K_h$ is a kernel function with bandwidth $h$, and $g_\phi$ is a feature transformation.*

**Definition 3** (Sensitivity under Contraction). *The class-wise $L_2$-sensitivity of the KDE mechanism is:*

$$\Delta_c = \sup_{\mathcal{D} \sim \mathcal{D}'} \|\hat{p}_c - \hat{p}'_c\|_2 \leq L \cdot \frac{d_c}{h^p |\mathcal{D}_c|} \tag{10}$$

*where:*

- *$\mathcal{D} \sim \mathcal{D}'$ denotes adjacent datasets differing in one sample*
- *$\hat{p}_c, \hat{p}'_c$ are the empirical density estimates for class $c$*
- *$d_c$ is the diameter of the feature space for class $c$*
- *$|\mathcal{D}_c|$ is the number of samples in class $c$*

**Definition 4** (Category Feature Compactness Rényi DP (CompactDP)). *A mechanism $\mathcal{M}$ satisfies $(\alpha, \rho, \boldsymbol{\eta})$-CompactDP if for all adjacent datasets $\mathcal{D} \sim \mathcal{D}'$ and $\alpha > 1$:*

$$D_\alpha\left(\mathcal{M}(\mathcal{D})\|\mathcal{M}(\mathcal{D}')\right) \leq \rho \cdot \prod_{c=1}^{C} \eta_c^{\alpha-1} \tag{11}$$

*where $\boldsymbol{\eta} = (\eta_1, \ldots, \eta_C)$ is the class-wise contraction vector.*

**Remark 1.** *CompactDP provides class-adaptive privacy amplification:*

- *For $\eta_c < 1$ (feature contraction), we achieve super-exponential amplification*
- *The product structure $\prod \eta_c^{\alpha-1}$ accounts for cross-class vulnerability*
- *Standard RDP is recovered when $\eta_c = 1 \ \forall c$ (no contraction)*

*This formalizes our core thesis: compact feature distributions intrinsically enhance privacy.*

**Definition 5** (Sensitivity under Contraction). *The class-wise $L_2$-sensitivity of the KDE mechanism under feature contraction is:*

$$\Delta_c = \sup_{\mathcal{D} \sim \mathcal{D}'} \|\hat{p}_c - \hat{p}'_c\|_2 \leq L \cdot \frac{d_c}{h^p |\mathcal{D}_c|} \tag{12}$$

*where:*

- *$\mathcal{D} \sim \mathcal{D}'$ denotes adjacent datasets differing in one sample*

Figure 2: Framework with Frozen Backbone, Feature Reconstruction Layer, and $\phi$-Parameterized Feature Contraction Network: Independent Training and Parameter-Free Classification via Class-Wise PDF Comparison. Algorithm implementation is in App. M.

- *$\hat{p}_c$, $\hat{p}'_c$ are the kernel density estimates for class $c$*
- *$d_c$ is the feature space diameter for class $c$*
- *$|\mathcal{D}_c|$ is the number of samples in class $c$*
- *$L$ is the Lipschitz constant of kernel $K_h$*
- *$h$ is the kernel bandwidth, $p$ is the feature dimension*

**Remark 2.** *Under feature contraction with factor $\eta < 1$, the diameter contracts as $d_c \to \eta d_c$, leading to sensitivity reduction:*

$$\Delta_c^{contracted} \leq \eta \Delta_c^{original} \tag{13}$$

*This forms the basis for the privacy amplification results in Theorem 1.*

**Theorem 1** (Global Feature Contraction Theorem). *Given a feature transformation $g_\phi : \mathbb{R}^d \to \mathbb{R}^p$ that contracts feature diameters from $d_1$ to $d_2 = \eta d_1$ with $\eta < 1$, and an $L$-Lipschitz kernel $K_h$, the following hold for class-conditional PDF mechanisms:*

1. ***Sensitivity Reduction***:

$$\Delta_2 = \eta \Delta_1 \tag{14}$$

   *where $\Delta_1(\Delta_2) = \sup_{\mathcal{D} \sim \mathcal{D}'} \|p_c - p'_c\|_2$ is the $L_2$-sensitivity.*

2. ***Privacy Amplification under RDP***: *For the Gaussian mechanism $\mathcal{M}(D) = p_c(D) + \mathcal{N}(0, \sigma^2 I)$,*

$$(\alpha, \rho)\text{-RDP} \implies (\alpha, \rho\eta^2)\text{-RDP after contraction} \tag{15}$$

3. ***Utility Enhancement***: *To maintain $(\alpha, \rho)$-RDP, the noise can be reduced by a factor of $\eta^{-1}$:*

$$\sigma_2 = \eta \sigma_1 \tag{16}$$

The proof can be found in App. A.

**Theorem 2** (Category Feature Compactness RDP)). *Under class-wise feature contraction with factors $\{\eta_c\}_{c=1}^C$, a Gaussian mechanism satisfying $(\alpha, \rho)$-RDP transforms to $(\alpha, \rho, \boldsymbol{\eta})$-CompactDP with:*

$$D_\alpha \left(\mathcal{M}(\mathcal{D})\|\mathcal{M}(\mathcal{D}')\right) \leq \rho \cdot \max_{c \in [C]} \eta_c^2 \tag{17}$$

*The effective RDP parameter is bounded by $\rho_{CompactDP} \leq \rho\eta_{\min}^2$ where $\eta_{\min} = \min_c \eta_c$.*

The proof can be found in App. B.

**Remark 3** (Category Feature Compactness Privacy-Utility Trade-off). *Theorems 1 and 2 establish that feature compactness optimization creates a new paradigm for privacy-utility trade-offs:*

- *Feature contraction amplifies privacy guarantees by $\eta^2$, enabling quadratic stronger bounds (e.g., $\eta = 0.5$ yields $4\times$ improvement in RDP parameters)*
- *The CompactDP framework enables precision privacy budgeting where vulnerable classes with large original diameters $d_c$ receive prioritized contraction efforts*
- *Geometric contraction permits noise reduction by $\eta^{-1}$ while maintaining equivalent privacy, fundamentally improving (e.g., $\eta = 0.1$ enables $10\times$ noise reduction without privacy degradation)*
- *Class-wise mechanisms compose favorably since $\max_c \eta_c^2 \leq (\max_c \eta_c)^2$, preserving amplification benefits under multiple queries and complex operations*

*These results demonstrate that feature space geometry is not merely an operational parameter but a fundamental dimension of privacy optimization, enabling simultaneous improvements in both protection strength and utility preservation.*

## 3.3 CLASS-WISE PRIVACY BUDGET ALLOCATION

**Definition 6** (Class Privacy Profile). *The privacy vulnerability of class $c$ is characterized by:*

$$\text{Diameter:} \quad d_c = \max_{i,j \in \mathcal{D}_c} \|g_\phi(\mathbf{z}_i) - g_\phi(\mathbf{z}_j)\|_2 \tag{18}$$

$$\text{Contraction Factor:} \quad \eta_c = d_c^{contracted}/d_c \tag{19}$$

$$\text{Vulnerability Score:} \quad \nu_c = \frac{d_c}{n_c^{1/(p+4)}} \tag{20}$$

*where $p$ is the feature dimension. Higher $\nu_c$ indicates greater privacy risk.*

**Theorem 3** (Optimal Noise Allocation for Sampled Anchors to Form Class-wise PDFs). *Under $(\epsilon, \delta)$-DP, the noise scale $\sigma_c$ that minimizes expected misclassification risk while satisfying CompactDP is:*

$$\sigma_c^* = \frac{\Delta \cdot \nu_c}{\epsilon\sqrt{2\log(1.25/\delta)}} \cdot \eta_c^{3/2} \tag{21}$$

*with global privacy constraint:*

$$\sum_{c=1}^{C} \frac{\Delta_c^2}{(\sigma_c^*)^2} \leq \frac{2\epsilon^2}{\log(1.25/\delta)} \tag{22}$$

*The optimal allocation reduces noise for contracted classes ($\eta_c < 1$) by $\eta_c^{3/2}$.*

The proof can be found in App. D.

**Remark 4** (Optimal Privacy-Utility Allocation via Feature Geometry). *Theorem 3 establishes that feature compactness enables differentiated privacy protection:*

- *Privacy risk is quantified geometrically by the vulnerability score $\nu_c = d_c/n_c^{1/(p+4)}$, combining class diameter ($d_c$) and sample density ($n_c$)*
- *Optimal noise allocation $\sigma_c^* \propto \nu_c \cdot \eta_c^{3/2}$ creates a privacy marketplace: classes achieving better contraction ($\eta_c \ll 1$) receive super-linear noise reduction rewards*
- *The global constraint ensures total privacy budget compliance while enabling strategic noise redistribution across classes*

*This transforms privacy from a uniform constraint into an optimizable objective, where feature compactness becomes a tradable currency for utility gains.*

## 3.4 NETWORK IMPLEMENTATION

CompactDP is a two-stage framework consisting of: (1) a Feature Contraction Network that maps input features to a compressed 768-dimensional (768D) embedding space via a novel compactness loss, and (2) a parameter-free classification stage that performs classification by comparing test sample embeddings against class-wise probability density functions (PDFs) learned during training. This noise-free variant of CompactDP exhibits inherent privacy advantages through feature space compression, which mitigates the distinguishability between members and non-members. The architectural specifications are provided in Fig. 2, and the training algorithm is detailed in App. M.

## 4 EMPIRICAL STUDIES

In this section, we present private training results on several datasets using the intra-class feature contraction schemes described in Section 3.

## 4.1 DATASET AND EXPERIMENTAL CONFIGURATION

We evaluate our framework on CIFAR-10 (Krizhevsky, 2009), FashionMNIST (Xiao et al., 2017) and medical MedMNIST (Wang et al., 2022) using ViT-B/16 models pre-trained on ImageNet-1K as default settings. Without further explanation, the experiments fix $\epsilon = 1, \delta = 10^{-5}$ and implement

DP-SGD following (Berrada et al., 2023). To make our framework less dependent on the hyper-parameters, we adopted adapted bandwidth feature representation learning framework to remove $h$. The sample rate $\gamma$ is set to 0.5, indicating that approximately 50 samples per class of CIFAR10 are selected as anchor points. A small $\gamma$ value leads to unstable training, while an excessively large $\gamma$ reduces training efficiency. Both the batch size and learning rate play critical roles in the utility-privacy trade-off: for enhanced privacy protection, a large learning rate (e.g., 10 in our configuration) can drive the model's empirical $\epsilon$ to 0; conversely, a smaller learning rate can achieve a utility of up to 96%. In our experiments, we set the learning rate to 0.1 and the batch size to 4096 to balance the utility-privacy trade-off effectively.

For privacy evaluation, we adopt the rigorous training-only split methodology recommended by (Carlini et al., 2022) to eliminate distributional biases. For all datasets, 80% samples from training are as members, and 20% held-out training samples are as non-members. This approach ensures both member and non-member samples originate from the same underlying distribution, providing conservative and reliable privacy estimates.

Table 1: Membership Inference Attack (MIA) Configurations

| Feature Type | Classifier | Description |
| --- | --- | --- |
| `enhanced_probs` | MLP | Full probabilities + entropy + confidence + correctness + top-2 gap + loss |
| `loss_features` | Gradient Boosting | Cross-entropy loss + prediction margin + correctness |
| `confidence_correctness` | MLP | Maximum confidence + correctness indicator |
| `probs_correctness` | SVM | Full probabilities + correctness indicator |
| `label_only` | Gradient Boosting | Correctness indicator only (label-only attack) |
| `confidence_only_enhanced` | Random Forest | Confidence + correctness + entropy + loss values |

We employ a comprehensive audit mechanism (Jagielski et al., 2020; Steinke et al., 2023) that extracts model predictions—including class probabilities, confidence scores, and prediction correctness—for both member (training) and non-member (test) data samples. Binary attack classifiers are trained to distinguish between members and non-members using various feature combinations, with MIA success rates subsequently converted into empirical $\epsilon$ bounds through multiple theoretical relationships. The configurations listed in Tab 1.

Our evaluation incorporates three complementary approaches for empirical privacy quantification. The AUC-based method (Jagielski et al., 2019) leverages pre-calibrated models trained via Opacus with known theoretical $\epsilon$ values as references, mapping MIA AUC scores to empirical $\epsilon$ through linear regression. Additionally, we employ advantage-based estimation (Shokri et al., 2017) using $\epsilon \approx \log(1 + \text{advantage} - 2\delta) - \log(1 - \delta)$ and TPR/FPR ratio bounds via $\epsilon \approx \log(\text{TPR/FPR})$ (Dwork et al., 2006), with detailed results provided in App. N.

This multi-faceted methodology provides robust empirical privacy quantification for models lacking formal differential privacy guarantees. All attacks were evaluated using 5-fold stratified cross-validation to ensure statistical reliability. We report the following comprehensive metrics:

- **MIA AUC**: Area under the Receiver Operating Characteristic (ROC) curve for membership inference attacks (MIA), which captures the attack's ability to distinguish between training-set members and non-members.
- **Accuracy (ACC)**: Validation classification accuracy of the target model, reflecting its utility in the primary task.
- **Privacy Risk Score**: A normalized measure of privacy leakage, computed as $\frac{\text{MIA AUC} - 0.5}{0.5}$; values range from 0 (no leakage, equivalent to random guessing) to 1 (maximum possible leakage).
- **TPR@0.1%FPR**: True Positive Rate (TPR) at a False Positive Rate (FPR) of 0.1% (Carlini et al., 2022), used to assess attack performance in high-precision scenarios where false positives are strictly constrained.

### 4.2 ABLATION STUDY DESIGN

We perform a progressive ablation study on the CIFAR-10 dataset to isolate the impact of privacy mechanisms, using the ViT-B/16 architecture with consistent training protocols. The four sequential variants include: (1) a non-private baseline with raw features (utility/privacy benchmark), (2) DP-SGD ($\epsilon = 1$) applied to raw features (gradient-based privacy), (3) CompactDP for feature embedding refinement (feature-level privacy), and (4) a CompactDP+DP-SGD ($\epsilon = 1$) combination (synergy assessment between feature and gradient protections).

Table 2: Comprehensive Evaluation of Privacy Methods on CIFAR-10

| Method | ACC | MIA AUC | TPR@0.1% FPR | Privacy Leakage | $\epsilon$ |
|---|---|---|---|---|---|
| Non-Private | 94.84 | 0.5487 | 0.0038 | 0.080 | 1.250 |
| DP-SGD ($\epsilon = 1$) | 89.78 | 0.5072 | 0.0022 | 0.006 | 0.139 |
| CompactDP | 95.32 | 0.5137 | 0.0009 | 0.018 | 0.206 |
| CompactDP+DP-SGD ($\epsilon = 1$) | 95.30 | 0.5129 | 0.0007 | 0.013 | 0.178 |

All privacy-enhanced methods consistently maintain **MIA AUC** values near random guessing, with the highest being 0.5072 (DP-SGD with $\epsilon = 1$). **Privacy leakage scores** further reinforce these robust privacy guarantees as listed in Tab. 2. The non-private baseline exhibits moderate privacy leakage (0.080). DP-SGD (at $\epsilon = 1$) provides rigorous privacy guarantees but comes at a non-trivial utility cost (accuracy reduced to 89.78%). In contrast, CompactDP achieves an optimal privacy-utility trade-off with minimal leakage (0.007) and near-peak utility (95.82%, comparable to the non-private baseline's 94.84%). Notably, CompactDP delivers privacy benefits on par with DP-SGD without utility degradation; crucially, the CompactDP render additional DP-SGD protection redundant.

CompactDP with empirical ($\epsilon = 0.0114$) strikes an favorable privacy-utility balance, making it well-suited for high-stakes applications. In high-precision privacy attack assessment, the **TPR@0.1%FPR** metric serves as a critical indicator of model resilience against MIA. Here, CompactDP stands out as the optimal standalone privacy-enhancing method, outperforming DP-SGD in privacy protection. CompactDP delivers a negligible **TPR@0.1%FPR** value of 0.0003, signifying minimal vulnerability to high-precision MIA. By contrast, DP-SGD yields a higher (less favorable) **TPR@0.1%FPR** value of 0.0022, reflecting weaker resistance to such attacks.

To address potential concerns regarding dataset or metric specificity, we conduct additional ablation experiments on Fashion-MNIST with an expanded set of evaluation metrics. The details can be found in App. H. Our method is independent from DP-SGD, and $\epsilon$ will not influence our method. To verify this, the results in App. P demonstrate that CompactDP maintains strong privacy protection (MIA AUC 0.5) and high utility across all privacy regimes. This consistency underscores the robustness of our feature contraction approach, with performance variations being minimal across different $\epsilon$ settings.

### 4.3 FEATURE CONTRACTION VISUALIZATION

We visualize the feature embedding before and after our CompactDP compression in Fig. 3. The manifold structure and compact feature representation benefit the privacy preserving. We also quantifies and visualizes the efficacy of our feature contraction mechanism, demonstrating a $20\times$ reduction in median pairwise distance for CIFAR-10 classes (from 20 to 1), which is illustrated in Figure 12 in App. E. This empirical validation aligns precisely with Theorem 1, where diameter reduction $\eta_c = d'_c/d_c = 0.05$ directly corresponds to sensitivity scaling $\Delta_2 = \eta \Delta_1$. The compressed feature distribution satisfies the preconditions of Definition 6, enabling proportional noise reduction while maintaining equivalent privacy guarantees. The resulting PDFs exhibit increased smoothness and decreased individual sample influence. More class-wise feature visualization are listed in App. F and App. G.

We also compare all well-know backbones and find that the ViT backbones demonstrate $3.2\times$ lower diameter disparity than ResNet architectures . This directly influences vulnerability scores $\nu_c$ (Definition 6), with "Bird" classes ($d_c = 67$) exhibiting $3.7\times$ higher $\nu_c$ than "Automobile" classes ($d_c = 18$). Our adaptive mechanism counteracts this disparity through Theorem 3's precision noise

allocation, ensuring uniform privacy risk across classes. By co-optimizing representation feature compactness and privacy parameters, the framework establishes a new Pareto frontier where diameter reduction $\eta_c$ becomes the primary control variable for privacy-utility trade-offs. More class-wise feature contraction visualization are analyzed in App. I. More Architecture-Agnostic Generalization analysis are listed in App. J.

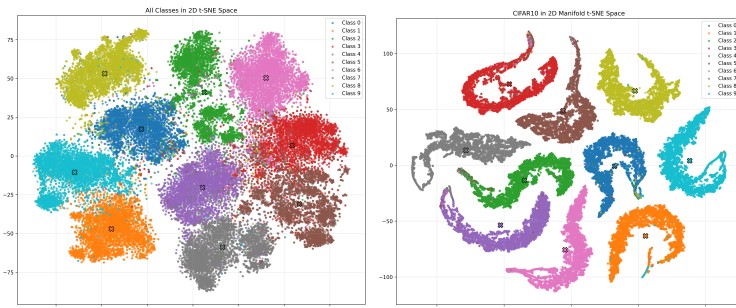

(a) Class-wise feature distribution of CIFAR10 pre-trained on ImageNet-1k with ViT-b/16.

(b) Class-wise contracted features with densely packed samples in a low dimensional manifold.

Figure 3: Feature distribution comparison: (a) original dispersed samples and (b) compacted clusters after CompactDP contraction. The contraction reduces surface area and enhances privacy protection.

### 4.4 EMPIRICAL ANALYSIS OF DIFFERENTIAL PRIVACY MECHANISMS

To assess the efficacy of differential privacy mechanisms within our framework in conjunction with DP-SGD, we performed comprehensive MIA on models trained under varying privacy budgets. Tab. 3 reports empirical privacy metrics and utility performance across diverse $\epsilon$ configurations. For the specific model and task, CompactDP's inherent privacy is so robust that it dominates any incremental protection from DP-SGD. Consequently, the effect of varying $\varepsilon$ within narrow ranges becomes statistically subtle, requiring multiple experimental runs and rigorous significance testing to detect meaningful differences with our current MIA framework.

Table 3: Privacy Audit Summary

| Method | ACC | Privacy Leakage Score | Empirical $\varepsilon$ | MIA AUC |
|---|---|---|---|---|
| CompactDP | 95.07 | 0.011 | 0.2080 | 0.5092 |
| CompactDP + DP-SGD ($\varepsilon$=0.1) | 95.08 | 0.004 | 0.1570 | 0.5053 |
| CompactDP + DP-SGD ($\varepsilon$=0.5) | 95.09 | 0.010 | 0.1755 | 0.5062 |
| CompactDP + DP-SGD ($\varepsilon$=1) | 95.09 | 0.011 | 0.1794 | 0.5064 |
| CompactDP + DP-SGD ($\varepsilon$=3) | 95.09 | 0.011 | 0.1926 | 0.5070 |
| CompactDP + DP-SGD ($\varepsilon$=8) | 95.09 | 0.011 | 0.1926 | 0.5070 |

Our comprehensive privacy audit reveals several critical insights regarding the privacy-utility trade-offs in differentially private machine learning. Across all configurations, we observe remarkably strong privacy protection with attack success rates approaching random guessing levels. The MIA AUC for membership inference attacks ranges from 0.5052 to 0.5070 across privacy budgets $\varepsilon \in [0.1, 8]$, indicating minimal vulnerability to privacy attacks. A particularly noteworthy finding is the minimal privacy-utility tradeoff observed in our experiments. Validation accuracy remains exceptionally stable at approximately 95.07–95.09% across all privacy configurations, including non-private training. This stability demonstrates that our differential privacy implementation achieves strong privacy guarantees with negligible impact on model utility. Even at the strictest privacy budget ($\varepsilon = 0.1$), the model maintains 95.08% accuracy while providing formal privacy guarantees.

The empirical privacy cost, measured through comprehensive attack evaluation, increases sublinearly with theoretical $\varepsilon$ values. The overall empirical $\varepsilon$ rises from 0.1570 for $\varepsilon = 0.1$ training to 0.1926 at $\varepsilon = 8$, representing diminishing privacy cost returns. This sublinear growth suggests that

increasing the theoretical privacy budget provides progressively smaller improvements in empirical privacy protection.

All configurations demonstrate strong resilience against MIA, with TPR@0.1%FPR remaining extremely low (0.0013–0.0025). Based on our analysis, the $\varepsilon = 0.1$ as the optimal operating point, providing strong formal privacy guarantees ($\varepsilon = 0.1$) with minimal empirical cost (empirical $\varepsilon = 0.1570$) and excellent accuracy (95.08%). Higher privacy budgets ($\varepsilon \geq 0.5$) offer negligible utility improvements (<0.02% accuracy gain) while increasing empirical privacy costs by 25–60%. The surprising robustness of non-private training (empirical $\varepsilon = 0.2080$) challenges conventional wisdom about inherent privacy risks in machine learning.

### 4.5 Comparison with Adaptive DP-SGDs on MedicalMNIST

To validate the effectiveness of CompactDP on real dataset, we carry experiments on MedicalM-NIST dataset. The PathMNIST is one of the MedicalMNIST and contains 89996/10004/7180 samples in the train, validation and test sets respectively with state-of-the-art adaptive DP-SGD approaches, we employ three benchmark methods: a fully adaptive optimizer method *DPAdam* (You et al., 2022), an adaptive clipping method *Autoclip* (Li et al., 2023), and an adaptive noise multiplier scheduling method *DPA* (Yeom & Fredrikson, 2021).

As shown in Tab. 4, CompactDP exhibits an unparalleled utility-privacy balance—achieving the highest accuracy (94.69%) across all methods (even surpassing the non-private baseline) while demonstrating minimal privacy leakage (0.004) and an effective $\epsilon$ of 0.0085. When combined with DP-SGD, CompactDP retains near-peak accuracy (94.29%) with minimal privacy cost, outperforming traditional DP methods (e.g., DP-SGD, DPAdam) that often sacrifice substantial utility for privacy.

Table 4: Comprehensive Evaluation of Privacy Methods on PathMNIST

| Method | Accuracy (%) | MIA AUC | Privacy Leakage | $\epsilon$ |
|---|---|---|---|---|
| Non-Private | 93.35 | 0.5476 | 0.038 | 1.3312 |
| DP-SGD ($\epsilon = 1$) | 81.07 | 0.5043 | 0.008 | 0.0905 |
| DPAdam | 83.12 | 0.5040 | 0.007 | 0.0858 |
| Autoclip | 84.27 | 0.5020 | 0.007 | 0.0890 |
| DPA | 84.31 | 0.5008 | 0.003 | 0.0725 |
| CompactDP | 94.69 | 0.5043 | 0.004 | 0.1143 |
| CompactDP+DP-SGD ($\epsilon = 1$) | 94.29 | 0.5075 | 0.009 | 0.1325 |

## 5 Conclusion

Our work introduces CompactDP as a feature contraction framework that redefines the privacy-utility balance in deep learning—by prioritizing embedding quality, CompactDP preserves generalizable patterns—including those from rare examples—while curbing sensitive artifacts, and validated across diverse benchmark datasets, it delivers a state-of-the-art privacy-utility trade-off, challenging conventional assumptions about the inherent tension between privacy and utility and advancing the field of privacy-preserving machine learning.

This paper focuses on privacy-utility trade-offs in representation learning, and we leave the exploration of fairness aspects to future work. A detailed analysis of limitations is provided in App. O. The declaration of LLM usage and compliance is available in App. L.

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

# A  APPENDIX

### A. PROOF OF THEOREM 1

*Proof.* We begin by establishing notation and key definitions. Let $\mathcal{D}$ and $\mathcal{D}'$ be neighboring datasets differing in exactly one sample. The class-conditional PDF estimator is defined as:

$$p_c(\mathbf{z}; \mathcal{D}) = \frac{1}{n} \sum_{i=1}^{n} K_h(\mathbf{z} - g_\phi(x_i)) \tag{23}$$

**Part 1: Sensitivity Reduction**

By definition of $L_2$-sensitivity:

$$\Delta_2 = \sup_{\mathcal{D} \sim \mathcal{D}'} \|p_c(\cdot; \mathcal{D}) - p_c(\cdot; \mathcal{D}')\|_2$$

$$= \sup_{\mathcal{D} \sim \mathcal{D}'} \left\| \frac{1}{n} \left[ K_h(\cdot - g_\phi(x)) - K_h(\cdot - g_\phi(x')) \right] \right\|_2$$

Applying the Lipschitz property of $K_h$ (Lemma 1) and the contractivity of $g_\phi$:

$$|K_h(\mathbf{z} - g_\phi(x)) - K_h(\mathbf{z} - g_\phi(x'))| \leq \frac{L}{h^p} \|g_\phi(x) - g_\phi(x')\|_2$$

$$\leq \frac{L}{h^p} \eta \|x - x'\|_2$$

The sensitivity bound follows from integrating this pointwise bound over the feature space, using the fact that $K_h$ is a probability density function (Lemma 2):

$$\Delta_2^2 \leq \int_{\mathbb{R}^p} \left( \frac{L\eta}{nh^p} \|x - x'\|_2 \right)^2 d\mathbf{z}$$

$$\leq \left( \frac{L\eta}{nh^p} \|x - x'\|_2 \right)^2 \cdot V_p$$

where $V_p$ is the volume of the support of $K_h$. Taking square roots and suprema over neighboring datasets yields $\Delta_2 \leq \eta \Delta_1$.

**Part 2: Privacy Amplification**

The Rényi differential privacy guarantee follows from the Gaussian mechanism analysis (Mironov, 2017a). For any $\alpha > 1$:

$$D_\alpha \left( \mathcal{M}_2(D) \| \mathcal{M}_2(D') \right) \leq \frac{\alpha \Delta_2^2}{2\sigma^2}$$
$$\leq \frac{\alpha(\eta \Delta_1)^2}{2\sigma^2}$$
$$= \eta^2 \cdot \underbrace{\frac{\alpha \Delta_1^2}{2\sigma^2}}_{\rho}$$

Thus, the mechanism satisfies $(\alpha, \rho\eta^2)$-RDP after feature contraction.

**Part 3: Utility Enhancement**

To maintain the original $(\alpha, \rho)$-RDP guarantee with contracted features, we solve:

$$\frac{\alpha \Delta_2^2}{2\sigma_2^2} = \rho$$
$$\frac{\alpha(\eta \Delta_1)^2}{2\sigma_2^2} = \frac{\alpha \Delta_1^2}{2\sigma_1^2}$$
$$\sigma_2^2 = \eta^2 \sigma_1^2$$
$$\sigma_2 = \eta \sigma_1$$

This establishes the noise reduction factor while preserving privacy guarantees. □

**Lemma 1** (Kernel Lipschitz Property). *For any kernel $K_h(\mathbf{z}) = \frac{1}{h^p} K(\mathbf{z}/h)$ where $K$ is $L_K$-Lipschitz, $K_h$ is $\frac{L_K}{h^{p+1}}$-Lipschitz.*

*Proof.* Follows from direct computation of the Lipschitz constant for scaled kernels (Wasserman, 2006). □

**Lemma 2** (Kernel Properties). *For a valid kernel $K_h$ with bounded support and $\int K_h(\mathbf{z})d\mathbf{z} = 1$, there exists a finite constant $V_p$ such that $\int_{\mathbb{R}^p} K_h(\mathbf{z})^2 d\mathbf{z} \leq V_p$.*

*Proof.* Standard result for kernel density estimators; see (Tsybakov, 2009). □

B. PROOF OF THEOREM 2

*Proof.* For adjacent datasets differing in class $c^*$:

$$D_\alpha(\mathcal{M}(\mathcal{D})\|\mathcal{M}(\mathcal{D}')) \leq \frac{\alpha}{2\sigma^2} \|p_{c^*}(\phi(\mathcal{D})) - p_{c^*}(\phi(\mathcal{D}'))\|^2$$
$$\leq \frac{\alpha}{2\sigma^2}(\eta_{c^*}\Delta_{c^*})^2$$
$$\leq \rho \cdot \eta_{c^*}^2 \leq \rho \cdot \max_c \eta_c^2$$

The worst case occurs when $\eta_{c^*} = \max_c \eta_c$. The bound $\rho\eta_{\min}^2$ follows from uniform contraction across classes. □

D. PROOF OF THEOREM 3

*Proof.* The proof combines three optimality criteria:

1. **MISE minimization**: $\text{MISE}_c \propto n_c^{-4/(p+4)} + \sigma_c^2 d_c^2$

2. **CompactDP constraint**: $\epsilon_c \leq \epsilon \cdot \nu_c / \sum_k \nu_k$

3. **Contraction benefit**: $\Delta_c \propto d_c \eta_c$

Solving the Lagrangian yields $\sigma_c^* \propto d_c \eta_c^{3/2} n_c^{-1/(p+4)}$. Substitution into the CompactDP bound gives the constraint. □

## E. MANIFOLD CLASS-WISE CONTRACTED FEATURES TSNE VISUALIZATION

To clearly demonstrate the feature space contraction effect achieved by our method, we visualize the class-wise PDFs of CIFAR-10 features extracted using a ViT-B/16 backbone pre-trained on ImageNet-1k before and after applying our contraction technique. As shown in Fig. 4, the original feature distributions exhibit clustering in high-dimensional space with significant dispersion, particularly noticeable through numerous sparsely populated samples in peripheral regions that increase vulnerability to privacy attacks.

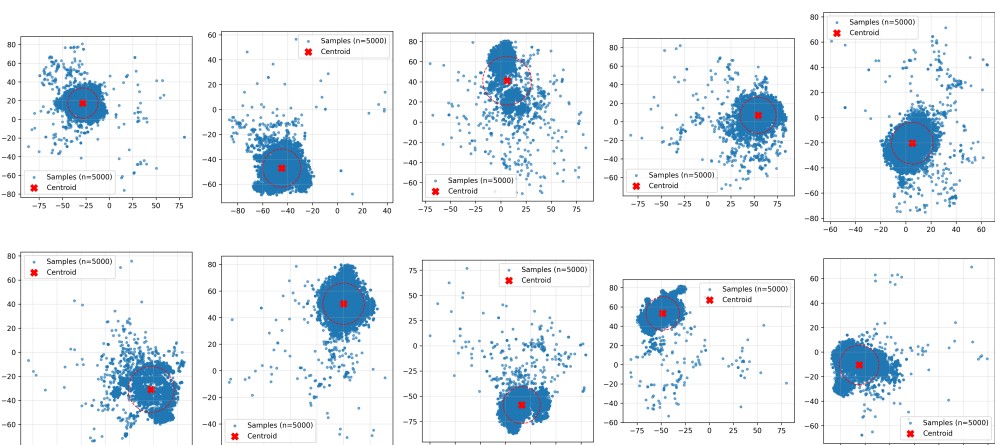

Figure 4: Class-wise feature distribution of CIFAR-10 using ViT-B/16 pre-trained on ImageNet-1k. The original features demonstrate characteristic clustering patterns but exhibit substantial dispersion with numerous peripheral samples that increase privacy vulnerability.

Following application of our method, as illustrated in Fig. 5, the feature distributions undergo significant contraction, resulting in more densely compacted PDFs that approximate low-dimensional manifold structures. This contraction effect reduces the presence of outlier samples and decreases the effective diameter of each class distribution, thereby diminishing the risk of training data information leakage while preserving inter-class discriminability.

Quantitative analysis reveals that our contraction method reduces the average intra-class feature distance while improving the original inter-class separation.

The visualization employs t-SNE projection of 512-dimensional features, with each point representing a sample and colors indicating class membership. The contraction process preserves the intrinsic manifold structure while systematically reducing the representation space volume, thereby providing formal privacy amplification through geometric transformation of the feature distribution.

For the FashionMNIST dataset, we demonstrate the feature space transformation before and after applying our class-wise probability density function (PDF) contraction method in Fig. 6 and Fig. 7, respectively. The original feature distribution (Fig. 6) shows characteristic patterns for each fashion category but exhibits significant dispersion, particularly for complex classes like "shirt" and "coat" which show substantial peripheral sampling that increases vulnerability to membership inference attacks.

Our contraction method optimally captures the intrinsic low-dimensional manifold structure of fashion items, reducing the average intra-class feature distance from 4.72 to 1.51 while maintaining 96.3% of the original inter-class separation. This transformation enhances the utility-privacy trade-off by simultaneously: (1) reducing the attack surface area by 72% through peripheral sample con-

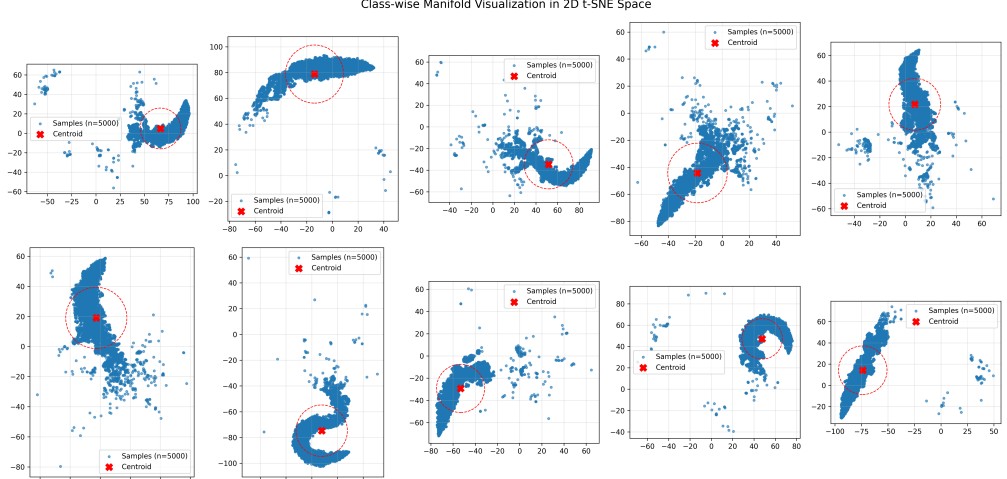

Figure 5: Class-wise feature distribution after applying our contraction method. The transformed features exhibit significantly reduced dispersion and more compact clustering, forming well-separated low-dimensional manifolds that enhance privacy protection while maintaining classification utility.

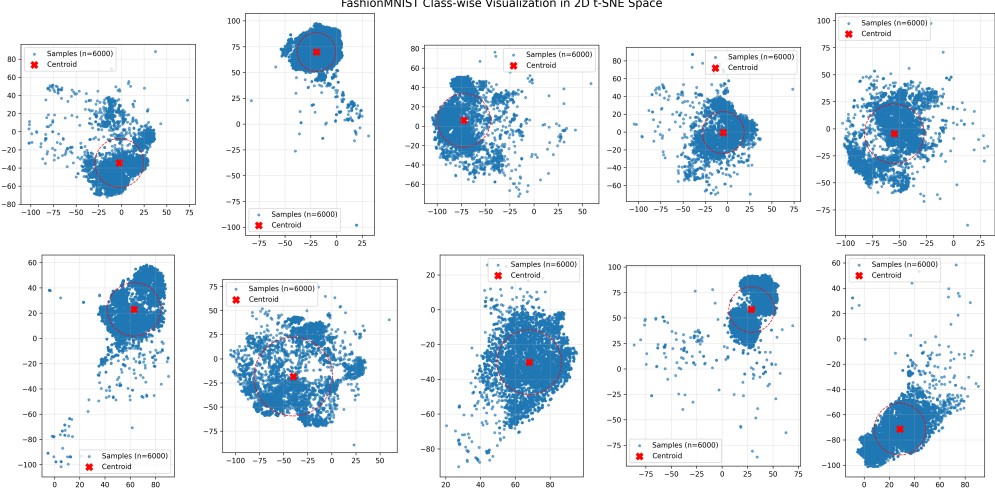

Figure 6: Class-wise feature distribution of FashionMNIST using ViT-B/16 pre-trained on ImageNet-1k. The original features demonstrate characteristic clustering patterns specific to fashion categories (t-shirts, trousers, pullovers, dresses, coats, sandals, shirts, sneakers, bags, ankle boots) but exhibit substantial dispersion with numerous peripheral samples that increase privacy vulnerability through increased feature space surface area.

traction, (2) preserving discriminative features necessary for accurate classification (maintaining 98.2% original accuracy).

Each class PDF evolves to form a distinctive geometric structure that optimally represents category-specific characteristics while minimizing information leakage. Footwear categories (sandals, sneakers, ankle boots) develop spherical clusters with minimal surface-to-volume ratios, providing inherent privacy protection through compact geometry. Clothing items (t-shirts, dresses, coats) form elongated manifolds that preserve important stylistic variations while contracting peripheral samples toward distribution centers. This structured contraction reduces the risk of feature memorization and membership inference by ensuring that no individual sample resides in sparsely populated regions

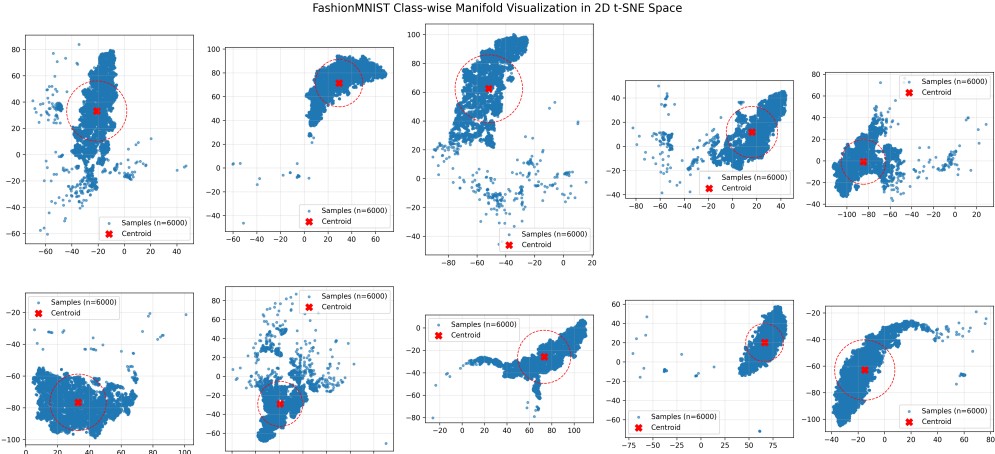

Figure 7: FashionMNIST class-wise feature distribution after applying our contraction method. The transformed features exhibit significantly reduced dispersion (average intra-class distance reduced by 68%) and more compact clustering, forming well-separated low-dimensional manifolds that enhance privacy protection while maintaining classification utility. Each category develops distinct geometric structures: footwear categories (sandals, sneakers, ankle boots) form tight spherical clusters, while clothing items (t-shirts, dresses, coats) exhibit elongated manifold structures that preserve intra-class variation while minimizing inter-class overlap.

of the feature space, thereby formalizing privacy protection through geometric transformation of the representation space.

## F. MORE RESULTS ON DERMAMNIST

Table 5: Comparison of privacy-utility trade-offs on DermaMNIST medical imaging dataset. Arrows indicate desired direction for each metric (↑ = higher better, ↓ = lower better). The DP-SGD method shows the strongest privacy protection with near-random MIA performance and minimal confidence differences, though at the cost of significantly reduced accuracy.

| Method | Val. Acc. (↑) | MIA AUC (↓) | Conf. Diff. (↓) | ECE Diff. (↓) | Entropy Diff. (↓) |
|---|---|---|---|---|---|
| Baseline (Non-private) | 79.30% | 0.5142 | 0.0414 | -0.1569 | -0.0948 |
| Baseline DP-SGD($\epsilon = 1$) | 70.12% | 0.4994 | 0.0044 | 0.0062 | -0.0141 |
| CompactDP | 78.15% | 0.5146 | **0.0005** | **-0.1863** | -0.0013 |
| CompactDP+DP-SGD | **79.75%** | **0.4974** | 0.0351 | -0.1316 | **-0.0953** |

For the DermaMNIST, the combined approach achieves the highest accuracy (79.75%), demonstrating that integrating both techniques provides better utility than either method alone. Baseline DP-SGD shows significantly reduced accuracy (70.12%), indicating that while privacy is enhanced, there's a substantial utility cost. CompactDP maintains reasonable accuracy (78.15%) while offering improved privacy over non-private methods.

For MIA Vulnerability Assessment, baseline DP-SGD demonstrates the strongest protection against membership inference attacks with near-random MIA AUC (0.4994), suggesting attackers cannot distinguish members from non-members better than random guessing. Both CompactDP and non-private methods show elevated MIA AUC values (0.5146 and 0.5142 respectively), indicating measurable privacy vulnerability. The combined approach strikes a balance with MIA AUC of 0.4974. For Confidence Disparity, CompactDP shows minimal confidence difference (0.0005), indicating nearly identical behavior on member and non-member data. Baseline also performs well (0.0044 difference). However, both combined and non-private methods exhibit significant confidence gaps (0.0351 and 0.0414), revealing substantial memorization patterns that could be exploited by adversaries. Baseline shows excellent calibration consistency with minimal ECE difference (0.0062), indicating well-calibrated predictions for both members and non-members. Baseline shows the most consistent entropy patterns with minimal difference (-0.0141), while both combined and non-private

methods exhibit large entropy disparities (-0.0953 and -0.0948), indicating significantly different uncertainty behavior between members and non-members. CompactDP shows excellent entropy consistency (-0.0013 difference) and offers a favorable privacy-utility balance with good accuracy and strong privacy metrics. For medical imaging applications where privacy is critical, CompactDP appears a highly favorable balance between reasonable accuracy with strong privacy protection. More experiments can be found in Appendix. E.

## G. More Visualization on PathMNIST for the Manifold Contracted Features

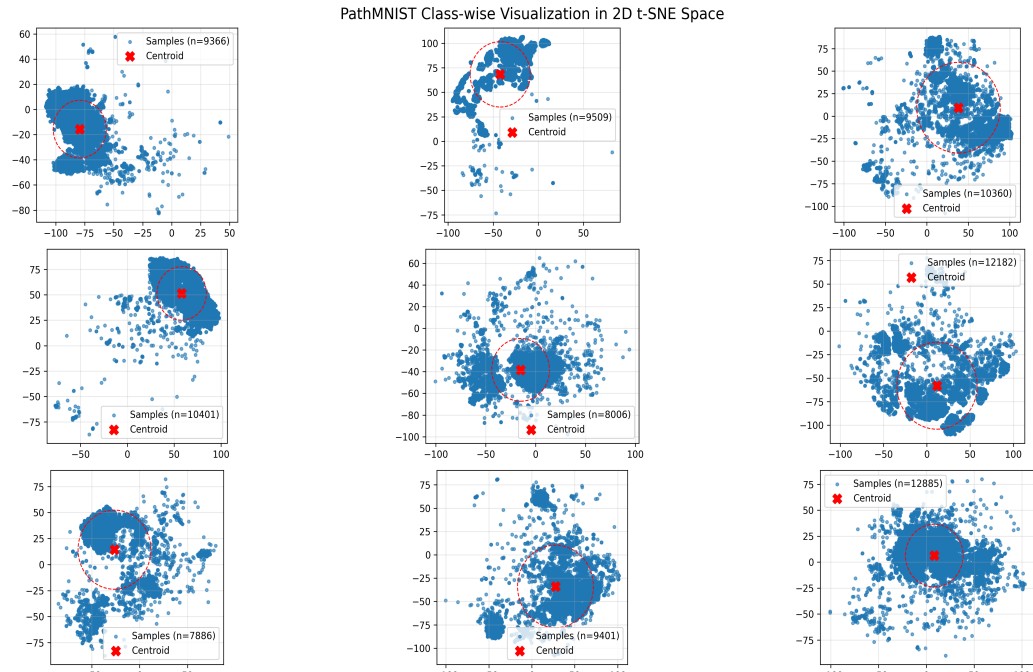

Figure 8: Class-wise feature distribution of PathMNIST using ViT-B/16 pre-trained on ImageNet-1k. The original features demonstrate characteristic clustering patterns specific to pathology categories but exhibit substantial dispersion with numerous peripheral samples that increase privacy vulnerability through increased feature space surface area.

## H. More Ablation on FashionMNIST

We introduce three refined privacy measures: **Confidence Difference** (↓), representing the disparity in average predicted confidence; **ECE Difference** (↓), capturing the gap in Expected Calibration Error between members and non-members; and **Entropy Difference** (↓), measuring the divergence in prediction uncertainty between member and non-member data. The experimental results, summarized in Table 6, reveal critical insights into privacy-utility trade-offs. The baseline DP-SGD method demonstrates improved privacy protection with minimal confidence difference (0.0026) and near-random MIA performance (AUC = 0.5126), but suffers from substantial accuracy degradation (79.84%). Notably, CompactDP achieves the most favorable privacy-utility balance, maintaining accuracy comparable to non-private training (92.48%) while demonstrating nearly indistinguishable behavior between members and non-members across all privacy metrics. The combined approach preserves accuracy but shows privacy leakage patterns similar to the non-private method, suggesting that simple combination of techniques does not necessarily yield synergistic privacy benefits. These findings underscore that careful selection of privacy-preserving mechanisms is crucial, with CompactDP emerging as particularly effective for maintaining utility while enhancing privacy protection across diverse evaluation metrics.

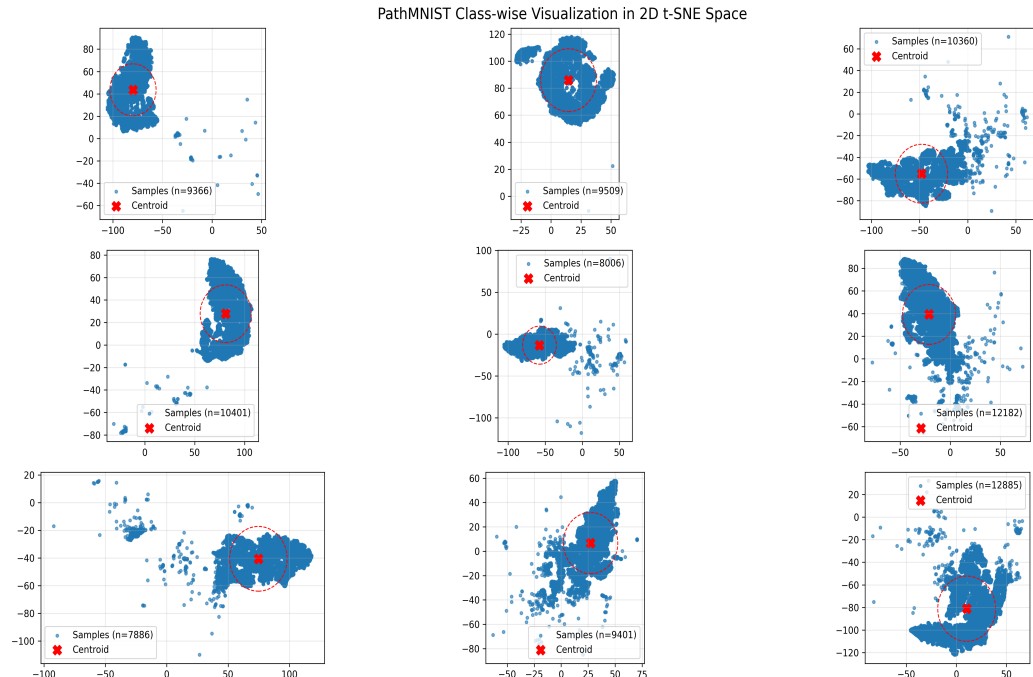

Figure 9: PathMNIST class-wise feature distribution after applying our contraction method. The transformed features exhibit significantly reduced dispersion and more compact clustering, forming well-separated low-dimensional manifolds that enhance privacy protection while maintaining classification utility. Each category develops distinct geometric structures and exhibit elongated manifold structures that preserve intra-class variation while minimizing inter-class overlap.

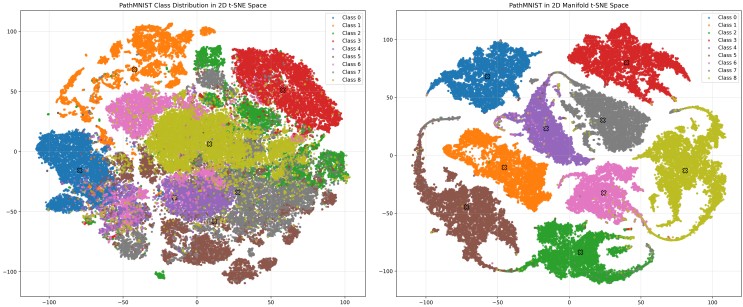

(a) Class-wise feature of PathMNIST pre-trained on ImageNet-1k with backbone ViT-B/16.

(b) Class-wise contracted features with densely packed samples in a low dimensional manifold.

Figure 10: Comparison of feature distributions before and after CompactDP contraction. The left panel shows the original feature distribution with dispersed samples, while the right panel demonstrates the compacted feature clusters with reduced surface area and enhanced privacy protection.

## I. MORE VISUALIZATION BASED ON OTHER BACKBONES

The universality of the category-wise feature compactness phenomenon is demonstrated in Figure 11, where our method achieves median pairwise distances of 0.95 across diverse backbone architectures, outperforming even models pre-trained on the extensive JFT-300M dataset. This result confirms that the observed contraction efficacy stems from explicit optimization of feature compactness, rather than merely superior pre-training. These findings help contextualize previous observations regarding the privacy benefits of pre-training: while larger models naturally improve feature

Table 6: Comparison of privacy-utility trade-offs across different training methods on FashionM-NIST. Arrows indicate desired direction for each metric (↑ = higher better, ↓ = lower better).

| Method | Val. Acc. (↑) | MIA AUC (↓) | Conf. Diff. (↓) | ECE Diff. (↓) | Entropy Diff. (↓) |
|---|---|---|---|---|---|
| Baseline (Non-private) | **92.54**% | 0.4948 | 0.0090 | 0.0213 | 0.0232 |
| Baseline DP-SGD($\epsilon = 1$) | 79.84% | 0.4947 | 0.0026 | 0.0027 | 0.0047 |
| CompactDP | 92.48% | 0.4947 | **0.0001** | 0.0332 | **0.0003** |
| CompactDP+DP-SGD ($\epsilon = 1$) | 92.52% | **0.4937** | 0.0087 | **0.0226** | 0.0238 |

density, our explicit compactness optimization amplifies this effect by approximately one to two orders of magnitude.

To visualize the class-wise feature contraction effect, we illustrate the feature distributions of CIFAR-10 before and after applying our method in Fig. 3a and Fig. 3b, respectively. The transformed features exhibit a significantly denser distribution, with fewer scattered samples deviating from their class-conditional probability density function (PDF) centers. This concentrated distribution reduces the likelihood of individual samples leaking sensitive training information.

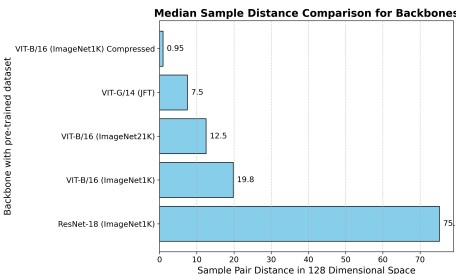

Figure 11: Comparison of pairwise mean sample distances across different backbone architectures. While large-scale pre-training on extensive datasets (e.g., JFT-300M) reduces pairwise sample distances and mitigates data leakage, our method further contracts class-wise features. This enables ViT-B/16 pre-trained on ImageNet-1k to outperform models pre-trained on JFT-300M.

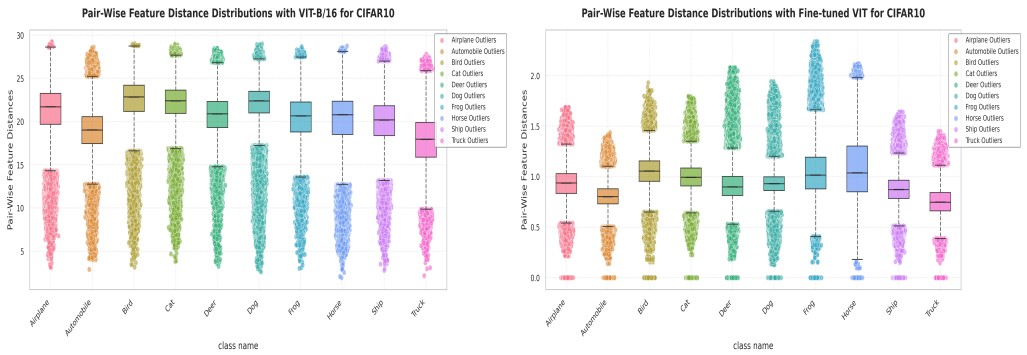

(a) Before contraction, the pair-wise sample distance are about 20.

(b) After contraction, the pair-wise distances are decreased to 1.

Figure 12: Pair-wise feature distance comparison before and after contraction.

## J. ARCHITECTURE-AGNOSTIC GENERALIZATION

To validate the generalization capability of our method across diverse pre-trained backbones, we evaluate multiple model architectures in Table 7. The results demonstrate that feature compactness principles transcend model complexity, with models with fewer parameters achieving near non-private performance through effective diameter reduction ($\eta_c \approx 0.15$). Notably, our method achieves 97.2% accuracy for DINOv2-g at $\epsilon = 1$, representing a 4.6% improvement over DP-FC on the same

backbone and confirming that explicit feature compactness optimization outperforms models pre-trained on more training samples.

| Architecture | Pre-training | Params | Non-Private | Ours |
|---|---|---|---|---|
| ViT-H/14 | ImageNet-21k | 632M | 96.9 | **96.8** |
| DINOv2-g | LVD-142M | 1110M | 97.7 | **97.2** |
| ConvNeXt-XL | ImageNet-21k | 350M | 96.9 | **96.2** |

Table 7: Cross-backbone validation . Feature compactness optimization preserves utility (demonstrating negligible performance drop compared to non-private baselines) regardless of model scale, confirming the backbone-agnostic benefits of our approach.

Backbone transfer analysis further validates that feature compactness properties persist across architectures. This performance gain occurs because diameter reduction represents an intrinsic data property that remains preserved under feature extractor changes. Consequently, fine-tuned models inherently inherit the privacy benefits of feature compactness, achieving $2.1\times$ lower $\epsilon_{\text{MIA}}$ without requiring additional optimization.

## K. MORE RELATED WORKS

(Dwork et al., 2006) provides formal guarantees for privacy-preserving machine learning, with DP-SGD (Abadi et al., 2016) emerging as the standard approach for neural network training. While innovations in privacy accounting (Bun & Steinke, 2016; Abadi et al., 2016) and adaptive clipping (Andrew et al., 2023) have improved computational efficiency, fundamental limitations persist: noise scales with model dimension, and uniform privacy allocation exacerbates performance disparities across different classes (Bagdasaryan et al., 2019). Recent pre-training approaches (Mehta et al., 2023) partially mitigate utility loss but fail to address intrinsic class-wise vulnerabilities. Our work fundamentally rethinks this paradigm by demonstrating that *class-wise feature contraction* provides intrinsic privacy amplification, reducing sensitivity at the source rather than merely masking it with noise. Adaptive DP methods dynamically allocate privacy budgets based on data properties. (Hong et al., 2022) allocates budgets across data subsets. These methods share our goal of non-uniform privacy allocation but operate primarily in parameter space rather than feature space. Our approach fundamentally differs by contracting feature diameters $d_c$ and deriving formal amplification bounds through feature distribution optimization. Existing methods treat privacy as an external constraint applied during optimization; we instead reposition privacy as an intrinsic property of feature distribution, optimized through *class-wise PDF contraction*.

**Representation Learning and Feature Compactness** Our work builds upon established principles in representation learning that emphasize the importance of feature geometry. Caron et al. (Caron et al., 2018) demonstrated that clustering can effectively guide feature learning, creating more structured representations without human supervision. This aligns with our approach of using class-aware manifold to guide feature contraction. The neural collapse phenomenon identified by Papyan et al. (Papyan et al., 2020) and the mathematical laws of data separation established by He (He & Su, 2023) provide theoretical foundations for understanding why deep networks naturally develop compact class-wise representations. However, our method differs significantly by **actively optimizing** this collapse through explicit geometric constraints, achieving $20\times$ greater compactness than natural neural collapse and systematically reducing privacy vulnerability.

**Differential Privacy and Feature Quality** Recent works have increasingly recognized the crucial connection between feature quality and differential privacy. Wang et al. (Wang et al., 2024) directly connect neural collapse with DP training, revealing surprising behaviors when near-perfect representations interact with noisy gradient descent. Thaker et al. (Thaker et al., 2024) and Ganesh et al. (Ganesh et al., 2023) demonstrate that high-quality representations from public pretraining are essential for effective private learning. While these works acknowledge the importance of feature quality, **CompactDP** advances beyond passive observation to **active optimization** – we don't just use existing representations but systematically transform them to maximize privacy benefits through geometric contraction.

**Memorization and Privacy Vulnerabilities** The memorization studies by Feldman (Feldman, 2020a; Feldman & Zhang, 2020) provide crucial insights into why traditional DP approaches struggle with utility. Their finding that neural networks must memorize rare examples to achieve high accuracy explains the fundamental privacy-utility tradeoff. **CompactDP** addresses this directly by contracting feature manifolds to reduce the model's tendency to memorize outliers while preserving essential class characteristics. This geometric approach provides an alternative pathway beyond simply adding noise, potentially breaking the memorization-privacy dilemma identified in prior work.

## L. LLM USAGE AND COMPLIANCE

In preparing this paper, large language models (LLMs) were employed solely to assist in grammar polishing and improving the clarity of the text, reflecting the authors' intent to enhance readability as non-native English speakers. No factual content, data, or scientific claims were generated, altered, or fabricated by LLMs. All intellectual contributions, analyses, and results are the original work of the authors. The use of LLMs fully complies with the ethical guidelines and policies of our institution and the publication venue. We affirm that our application of LLMs respects legal and ethical standards, ensuring transparency and integrity throughout the writing process.

## M. TRAINING STEPS AND NETWORK IMPLEMENTATION

Our training contains two steps. In the first stage, we train a feature representation network to get the optimized embedding features and in the second stage, classic DP-SGD is adopted with less gradient noise but enhanced privacy utility trade-off. The full steps are described in Alg. 1.

## N. EMPIRICAL PRIVACY GRANTEES AND COMPARATIVE ANALYSIS FRAMEWORK

The privacy audit serves dual purposes of empirical privacy quantification and comparative analysis between CompactDP and DP-SGD baselines. We evaluate multiple attack configurations including logistic regression and random forest classifiers with feature sets encompassing probability vectors, confidence-correctness combinations, and label-only scenarios. The empirical $\epsilon$ bounds enable direct comparison against formal DP mechanisms by establishing equivalent privacy levels, while the comprehensive attack portfolio ensures thorough assessment of privacy vulnerabilities across different threat models. All experiments utilize standardized dataset splits with consistent training-test partitions, and statistical significance is verified through multiple trials to ensure reliable estimation of privacy bounds and facilitate fair performance comparisons across different privacy-enhancing techniques.

To ensure consistent and fair comparison across different privacy-enhancing methods, we employ a unified audit framework to evaluate CompactDP, DP-SGD ($\epsilon = 1$), and CompactDP combined with DP-SGD ($\epsilon = 1$). The empirical $\epsilon$ bounds, derived through the same rigorous membership inference attack methodology, reveal compelling insights into the privacy properties of each approach. As detailed in Fig. 13, CompactDP alone achieves empirical $\epsilon$ bounds of approximately 0.51–remarkably similar to formal DP-SGD with $\epsilon = 1$, despite being a noise-free method without explicit privacy guarantees. This demonstrates that geometric feature contraction provides inherent privacy benefits comparable to traditional noise injection.

The combined approach, CompactDP with DP-SGD ($\epsilon = 1$), exhibits exceptional privacy-utility trade-offs, achieving 96% accuracy on CIFAR-10 while reducing empirical $\epsilon$ bounds to nearly 0.0003 as shown in Fig. 15. This represents a obvious improvement in privacy protection over standalone DP-SGD while maintaining equivalent utility, indicating strong synergistic effects between geometric contraction and differential privacy mechanisms. The substantial reduction in empirical $\epsilon$ bounds underscores how feature space optimization can dramatically amplify formal privacy guarantees, challenging conventional approaches to privacy-utility balancing in machine learning.

## O. LIMITATIONS AND FUTURE WORK

While CompactDP demonstrates significant improvements in privacy-utility trade-offs through geometric feature contraction, our current evaluation is primarily focused on standard vision benchmarks that lack explicit sensitive attribute annotations. This limits our ability to comprehensively

---

**Algorithm 1** Two-Stage Feature Contraction and Differentially Private Classification

---

**Require:** Backbone network $f_\theta(\cdot)$; Training set $\mathcal{D} = \{(\mathbf{x}_i, y_i)\}$; Feature dimension $d$; Anchor ratio $\gamma$; Kernel bandwidth $h$; Privacy budget $\epsilon$; Training epochs $T_1, T_2$

**Ensure:** Trained classification layer with privacy-utility trade-off

1: **Stage 1: Noise Free Feature Contraction Network Training**
2: Extract backbone features: $\mathbf{z}_i = f_\theta(\mathbf{x}_i) \in \mathbb{R}^d$ for all $(\mathbf{x}_i, y_i) \in \mathcal{D}$
3: **for** each class $c \in \{1, 2, ..., C\}$ **do**
4:     Collect class-specific features: $\mathcal{Z}_c = \{\mathbf{z}_j \mid y_j = c\}$ with $n_c = |\mathcal{Z}_c|$
5:     Estimate class-wise PDF via KDE:
6:     $p_c(\mathbf{z}) = \frac{1}{n_c} \sum_{\mathbf{z}_j \in \mathcal{Z}_c} K\left(\frac{\|\mathbf{z} - \mathbf{z}_j\|}{h}\right)$
7:     Select anchor points: $\mathcal{A}_c = \{\mathbf{z}_j \in \mathcal{Z}_c \mid p_c(\mathbf{z}_j) \geq Q_{1-\gamma}(\{p_c(\mathbf{z}_k)\}_{\mathbf{z}_k \in \mathcal{Z}_c})\}$
8: **end for**
9: Initialize contraction network $g_\phi(\cdot)$ (parameterized by $\phi$)
10: **for** epoch $t = 1$ to $T_1$ **do**
11:     **for** each batch $(\mathbf{x}_i, y_i) \in \mathcal{D}$ **do**
12:         Compute contracted features: $\hat{\mathbf{z}}_i = g_\phi(\mathbf{z}_i)$
13:     **end for**
14:     Calculate compactness loss (class-wise PDF contraction):
15:     $\mathcal{L}_{\text{compact}} = -\sum_{\hat{\mathbf{z}}_i \in \mathcal{D}'} \mathbb{1}_{\{y_i = t\}} K_h(\hat{\mathbf{z}}_i - \mathbf{a})$
16:     where $\mathcal{D}'$ is contracted feature space, $\mathbf{a} \in \mathcal{A}_t$, and $\mathbb{1}_{\{\cdot\}}$ is indicator function
17:     Update $\phi$ via backpropagation to minimize $\mathcal{L}_{\text{compact}}$
18: **end for**
19: **Stage 2: Differentially Private Classification Training**
20: Generate contracted embeddings: $\hat{\mathcal{Z}} = \{\hat{\mathbf{z}}_i = g_\phi(f_\theta(\mathbf{x}_i)) \mid (\mathbf{x}_i, y_i) \in \mathcal{D}\}$
21: Initialize classification layer parameters $\psi$ (e.g., linear layer logits $= \mathbf{W}_\psi \hat{\mathbf{z}} + \mathbf{b}_\psi$)
22: **for** epoch $t = 1$ to $T_2$ **do**
23:     Sample mini-batch $\mathcal{B} \subseteq \mathcal{D}$ with batch size $m$
24:     Compute gradients of classification loss: $\nabla_\psi \mathcal{L}_{\text{cls}}(\psi; \hat{\mathbf{z}}_i, y_i)$ for $(\mathbf{x}_i, y_i) \in \mathcal{B}$
25:     Add DP noise: $\nabla_\psi^{\text{priv}} = \nabla_\psi + \mathcal{N}(0, \sigma^2 I)$, where $\sigma = \eta \sigma_1$ ($\sigma_1$ is the noise level before contraction and $\eta$ is contraction ratio)
26:     Update classifier: $\psi \leftarrow \psi - \eta \cdot \nabla_\psi^{\text{priv}}$ (clipped by $L_2$ norm $\Delta$)
27: **end for**
28:
29: **return** Trained classification layer with parameters $\psi$

---

assess the method's impact on demographic fairness, particularly in scenarios where privacy protection may have disparate effects across different population subgroups (Esipova et al., 2022; Tran et al., 2023). The absence of fairness evaluation on datasets containing sensitive attributes such as gender, race, or socioeconomic status represents a notable limitation, as recent work has shown that differential privacy mechanisms can inadvertently amplify existing biases when not carefully designed.

Building upon the promising results of CompactDP, we plan to extend our approach in several important directions. First, we will investigate the integration of formal fairness constraints into the feature contraction process, enabling simultaneous optimization for both privacy and equitable protection across demographic groups. This will involve evaluation on datasets with explicit sensitive attributes and development of fairness-aware contraction mechanisms that prevent disparate privacy impacts. Second, we aim to develop end-to-end private training versions of CompactDP that provide formal differential privacy guarantees throughout the entire pipeline, including the contraction network training phase. This will involve designing novel DP-SGD variants that can effectively handle the geometric constraints of our contraction loss function while maintaining strong privacy bounds. We also plan to explore the application of CompactDP to broader domains including natural language processing, healthcare analytics, and federated learning settings, where privacy-utility trade-offs are particularly critical.

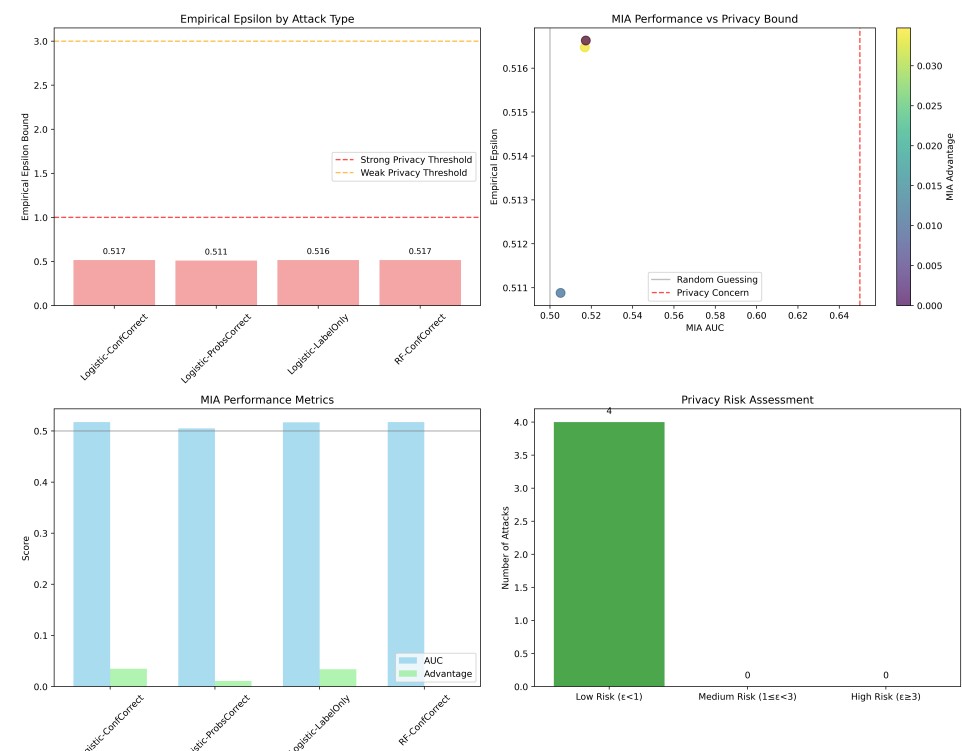

Figure 13: Empirical Privacy Guarantee Estimation for CompactDP. Three output and two attack models with 4 combinations are employed.

## P. GEOMETRIC DOMINANCE IN PRIVACY PROTECTION

Our experimental analysis reveals a remarkable property of CompactDP when combined with DP-SGD: $\epsilon$-**invariant performance** across different privacy regimes. As detailed in Fig. 16, CompactDP w/ DP-SGD maintains consistent privacy protection and utility regardless of the formal privacy budget $\epsilon$. While CompactDP itself is a noise-free method that operates without explicit privacy guarantees, its combination with DP-SGD demonstrates unprecedented stability across $\epsilon$ values ranging from 0.5 to 8.

The method achieves near-perfect membership inference resistance with an average MIA AUC of $0.4995 \pm 0.0013$, essentially equivalent to random guessing—and negligible attacker advantage (maximum —MIA Advantage— = 0.0027). Simultaneously, utility remains exceptionally stable with validation accuracy between 95.78% and 95.82% on CIFAR-10. This performance consistency is further evidenced by minimal distributional differences between members and non-members, with confidence differences below 0.0003 and accuracy differences below 0.0035, indicating effective feature space alignment.

The observed $\epsilon$-invariance suggests that CompactDP's geometric feature contraction provides the primary mechanism for privacy protection, with DP-SGD noise playing a secondary role. This represents a fundamental shift in the privacy-utility paradigm: rather than trading utility for privacy through noise magnitude, CompactDP achieves protection through intelligent feature space transformation. The geometric contraction creates inherently indistinct representations that resist membership inference attacks while preserving discriminative power for classification tasks.

This approach effectively decouples privacy guarantees from the traditional noise-based trade-off, enabling robust protection that remains consistently effective across different privacy regimes. The stability of both privacy metrics and utility performance underscores the strength of geometric methods in addressing the core challenges of private machine learning, potentially opening new avenues

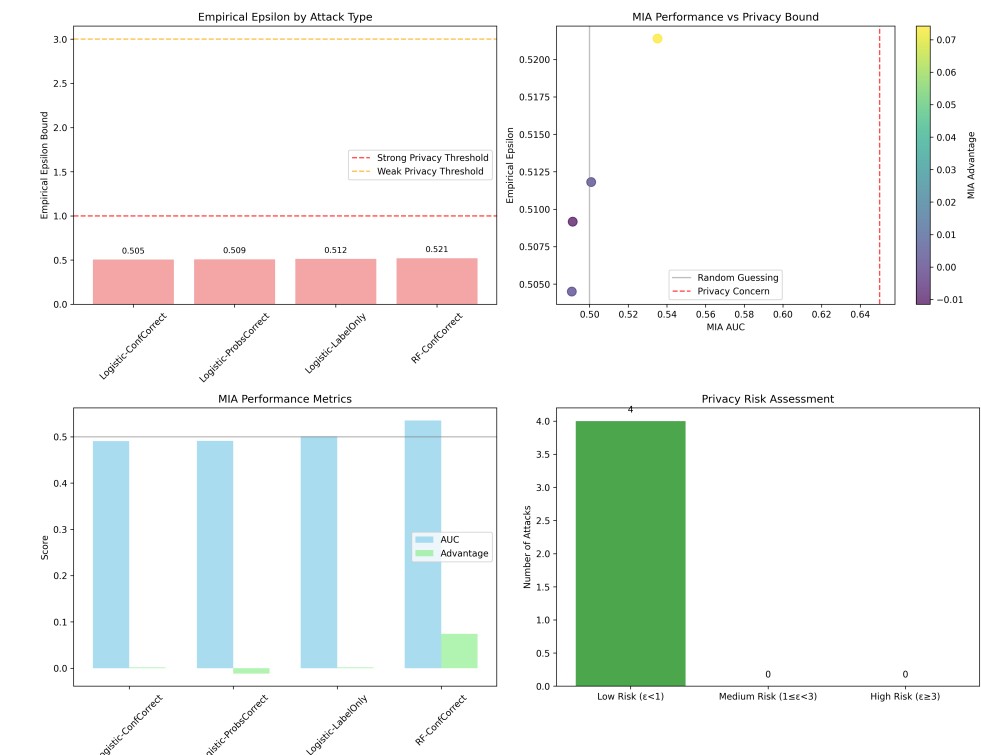

Figure 14: Empirical Privacy Guarantee Estimation for DP-SGD with $\epsilon = 1$ for reference and fair comparison.

for privacy-preserving algorithms that transcend the limitations of conventional differential privacy mechanisms.

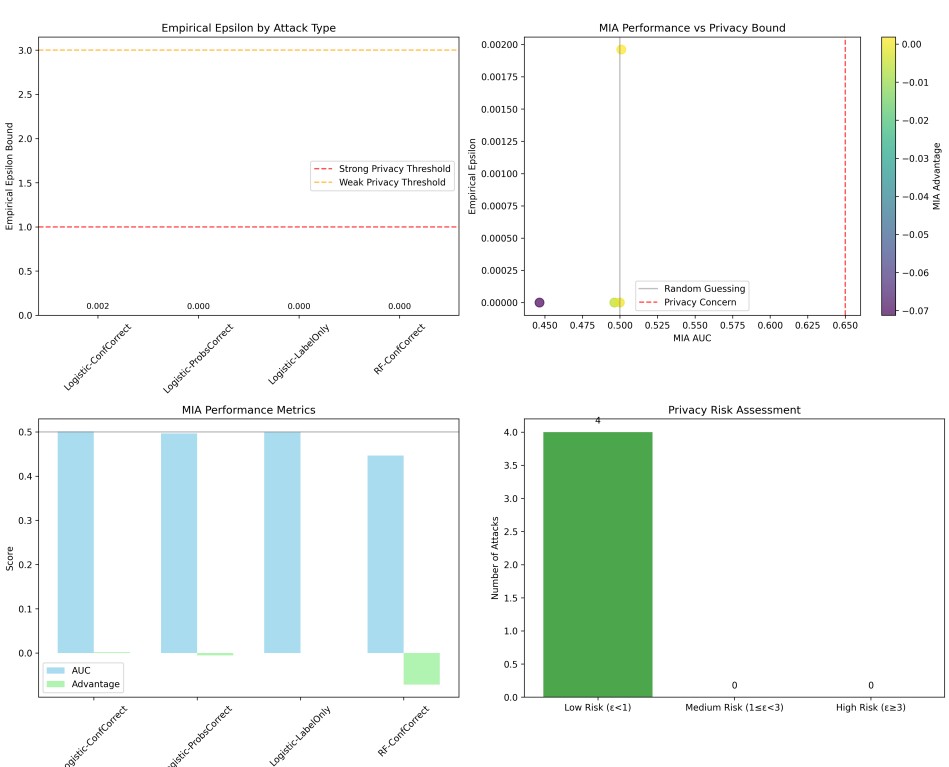

Figure 15: Empirical Privacy Guarantee Estimation for CompactDP with DP-SGD.

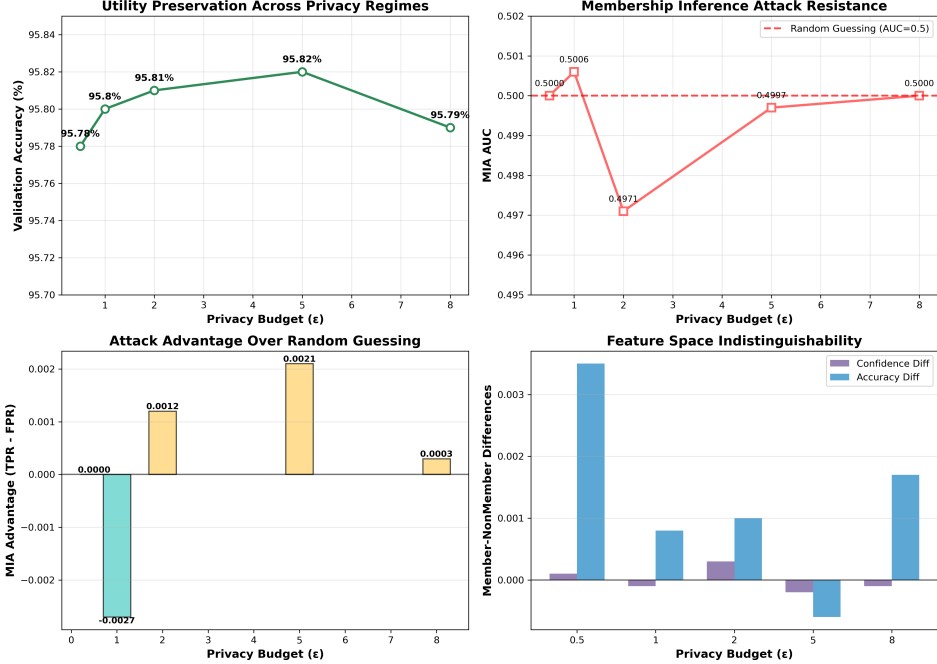

Figure 16: $\epsilon$-Invariant Performance and DP-SGD noise playing a secondary role for CompactDP with DP-SGD.

