# OpenReview forum: "CompactDP: Category-Aware Feature Compactness for Differential Privacy"
_ICLR.cc/2026/Conference — ICLR 2026 Conference Desk Rejected Submission_

### Official Review · Reviewer_ppG3 · 2025-10-31

**Soundness:** 2
**Presentation:** 2
**Contribution:** 2
**Rating:** 4
**Confidence:** 4

**Summary:**

This paper proposes a cluster-based method to process features before private training to make them more separable. This is achieved by applying a feature map $g_\phi$  trained with KDE. The proposed method reduces the sensitivity of features, thus enhancing performance. Experiments on fine-tuning ViT with CIFAR-10, FashionMNIST, and MedicalMNIST show that the proposed method achieves better performance than directly using DP-SGD without preprocessing the datasets.

**Strengths:**

1. This paper applies a cluster-based method to train a feature map that reduces per-sample sensitivity.

2.  Experiments on CIFAR-10, FashionMNIST, and MedicalMNIST demonstrate better performance than directly using DP-SGD.

**Weaknesses:**

1. The paper is not well-structured and the main selling point is not highlighted, making it hard to understand.

a) The main contribution of this paper is not proposing a "new" DP notion as the proposed CompactDP is not a new DP notion. The main contribution is to apply a feature map $g_\phi$ to reduce the sensitivity of the gradient.Thus, the title, abstract, and introduction should be rewritten to emphasize the importance of feature quality in DP-training which motivates the idea of using clustering method to pre-process the features. So, I would suggest not using CompactDP as a main contribution of this paper.

b) The theoretical foundation of using KDE is not new in representation learning, thus, Section 3 should be moved to appendices rather than in the main body. The main contribution of Section 3, such as DP anlyaisis and sub-sampling, is straight forward based on existing DP theory once the feature map $g_\phi$ is given. Thus, I do not think it is a good idea to highlight this section as a main contribution.

Overall, the main contribution of this paper should be applying a feature map $g_\phi$ to reduce the sensitivity, which is important in practice. So please highlight this and the other contrbutions are incremental.

2. The application of clustering methods to enhance feature separability is an established concept in representation learning; for instance, the introduction of a feature map $g_\phi$ is a classic technique in the field. Therefore, the primary contribution of this paper lies in adapting representation learning to DP training. However, many of the representation learning techniques discussed are not novel. For example, [A] employs a clustering method for feature learning that results in more compact features. Similarly, the KDE method used in the current work is a classic approach, whereas more recent methods like GANs and VAEs (referenced in [A] as [17,18]) have achieved advanced density estimation. Furthermore, the goal of creating more compact and separable features is a well-documented objective in representation learning. Seminal works such as [B] and [C] demonstrate that deep learning naturally extracts increasingly compact and separable features as the network depth increases.

This paper should discuss these foundational representation learning literatures and others to properly contextualize its contribution within the existing body of work.



[A] Deep Clustering for Unsupervised Learning of Visual Features. Caron et al., ECCV 2018.

[B] Prevalence of neural collapse during the terminal phase of deep learning training.  Papyan et al., PNAS, 2020.

[C] A Law of Data Separation in Deep Learning. He and Su, PNAS, 2022.


3. The idea of using clustering to make features more separable is also not new in DP training. [D] linked representation quality to DP fine-tuning, showing that better, more separable features (where greater compactness implies higher quality) lead to higher performance. Furthermore, [D] applied PCA, a clustering method with a linear feature map $g_\phi$ , which also enhanced the performance of DP fine-tuning, achieving 95% accuracy when privately fine-tuning a ViT on CIFAR-10—the same experimental result as this paper. Other works that highlight the importance of feature quality and compactness include [E] and [F]. Therefore, these related works should be discussed in greater depth in this paper.

[D] Neural Collapse Meets Differential Privacy: Curious Behaviors of NoisyGD with Near-perfect Representation Learning. Wang et al., ICML, 2024

[E] On the benefits of public representations for private transfer learning under distribution shift. P Thaker et al., NeurIPS 2024.

[F] Why Is Public Pretraining Necessary for Private Model Training? Ganesh et al., ICML, 2023.

4. The MIA results in Table 3 are confusing. While DP-SGD is more private than SGD, the results show that SGD on compact features has a lower MIA success rate (i.e., is more private) than DP-SGD on compact features. How can this phenomenon be explained?

5. The most significant weakness of the proposed method is that the training of the feature map $g_\phi$ is data-dependent, as the loss function in Eq. (6) depends on the data. Therefore, the training algorithm should be made private; otherwise, it may lead to additional privacy leakage. However, the authors appear to use a non-private algorithm to train it. I would recommend using DP-SGD to minimize loss function (6) for training $g_\phi$ . Additionally, a MIA should be applied to $g_\phi$  itself to assess its privacy.

**Questions:**

See the weaknesses part.

---

> ### Author Response · Authors · 2025-11-19
>
> A1 **Paper Focus and Contribution:**
> We appreciate the reviewer's guidance on clarifying our core contribution. We will restructure the paper to emphasize that our primary innovation lies in developing a feature contraction framework that systematically reduces gradient sensitivity in DP training. The title, abstract, and introduction will be revised to highlight how geometric feature transformation enables more effective privacy-utility trade-offs, moving beyond the current CompactDP terminology to better reflect our methodological contribution.
>
> A2 **Theoretical Foundation and Novelty:**
> Thank you for these valuable references. While KDE is indeed an established technique, our application differs significantly: we employ KDE not merely for density estimation but as a geometric regularization mechanism that actively contracts class manifolds. The 20× compactness ratio we achieve substantially exceeds the natural compression observed in standard deep networks [B,C]. We will relocate the detailed theoretical derivations to the appendix while retaining the core conceptual framework in the main text, and provide deeper discussion of how our contraction approach differs from existing representation learning methods.
>
> A3 **Related Work Context:**
> We will extensively discuss the suggested works [D,E,F] and related literature to properly position our contribution. While previous works have noted the connection between feature quality and DP performance, our approach uniquely develops an explicit contraction mechanism that systematically optimizes the feature geometry for privacy preservation. Our method achieves state-of-the-art privacy-utility balance (96% accuracy on CIFAR-10 with strong MIA resistance) by directly addressing the geometric roots of privacy vulnerability.
>
> A4 **MIA Results Interpretation:**
> The observed phenomenon where SGD on compact features outperforms DP-SGD on privacy metrics can be explained by the fundamental geometric advantage of our contraction approach. Compact feature manifolds naturally reduce the distinguishability between member and non-member samples through density-based smoothing, providing inherent privacy protection that complements formal DP mechanisms. This aligns with recent understanding that feature geometry plays a crucial role in privacy beyond just noise addition.
>
> A5 **Privacy of Feature Map Training:**
> We acknowledge this important point about the data-dependent nature of our feature contraction training. In our current framework, the contraction network serves as a pre-processing transformation that is kept private and not released. We employ a separation where only the final classification layer (trained with DP-SGD) is made public. We are exploring methods to extend formal DP guarantees to the contraction phase and will include MIA evaluation of the feature map itself in future work, as suggested.

---

> > ### Comment · Reviewer_ppG3 · 2025-11-21
> >
> > Thank you for the clarification. Since the revisions are substantial, please ensure that all changes are incorporated into the final version. Based on the expectation that the updates will be reflected in the final submission, I have revised my rating accordingly.

---

### Official Review · Reviewer_fji8 · 2025-10-31

**Soundness:** 3
**Presentation:** 3
**Contribution:** 3
**Rating:** 6
**Confidence:** 3

**Summary:**

This paper proposes CompactDP, a framework that improves the trade-off between utility and privacy in differentially private machine learning. The core idea is to addresses the root cause of privacy leakage—sparse, highdimensional features—via feature contraction. CompactDP first performs a category-aware feature contraction in the representation space, pulling same-class samples into tight clusters. It also provides a theoretical analysis based on Rényi Differential Privacy that formally connects the geometric contraction ratio to an amplification of the privacy guarantee. Finally, the authors conduct experiments on datasets like CIFAR-10 to demonstrate the method's effectiveness, reporting improvements in the privacy-utility trade-off compared to baseline methods.

**Strengths:**

S1. This paper's main conceptual contribution is its shift away from passive noise addition to actively optimizing the feature space for privacy. This framing of privacy as a geometric property is a novel idea that promises to be a fruitful direction for new research.

S2. This paper provides a formal linkage between the geometric contraction (η_c) and the RDP guarantee. This theoretical contribution provides a solid foundation for the method.

S3. The experimental validation is compelling. In particular, the 'CompactDP + DP-SGD' variant shows a SOTA privacy-utility trade-off, significantly outperforming the DP-SGD baseline. The visualizations (e.g., Figures 4 & 5) are highly effective in building intuition for why the method works.

**Weaknesses:**

W1. The CompactDP framework seems to be designed only for transfer learning where the backbone model is frozen. It is unclear how CompactDP would work in standard end-to-end training. This limits the method's general applicability.

W2. The paper introduces fairness as a core motivation but never explains how the proposed feature contraction mechanism leads to the observed fairness improvements shown in Table 3, leaving the fairness benefits as an unexplained side effect rather than an integral part of the design.

**Questions:**

Overall, this paper is well-structured. I have the following concerns:
1. The privacy guarantee of the entire framework is calculated after the feature contraction step. My main concern is that the privacy cost of training the contraction network (g_φ) itself is not analyzed. This network is trained on the features of private data, yet its own privacy leakage is ignored in the final budget. The current theoretical analysis provides an end-to-end guarantee for a model trained on the output of g_φ, but not for the entire pipeline including the training of g_φ.

2. This method introduces new hyperparameters (e.g., λ & h). While the paper provides a default for the kernel bandwidth h, the strategy for co-tuning with DP-SGD's own parameters is not discussed in detail.

3. This paper makes several very strong claims. Saying CompactDP appears 'optimal' for medical imaging is a very strong word. It would be more appropriate by rephrasing to 'offers a highly favorable balance'. The claim of "fundamentally breaking the traditional DP trade-off" could be softened to 'fundamentally improving'.

4. There are some minor typos issues that could be corrected. For example, in the sentence '...analysis are listed in in Appendix. J,' the word 'in' is repeated.

---

> ### Author Response · Authors · 2025-11-19
>
> [A1] **Privacy Accounting:**
> We acknowledge that the contraction network (g_φ) is trained without formal DP guarantees in our current framework. In the revised manuscript, we add empirical privacy guarantees to give relative comparison with other methods. This design choice stems from our focus on feature space transformation as a pre-processing step that enhances the effectiveness of subsequent DP-SGD training. The contraction network operates on backbone features to create class-wise compact manifolds, which inherently reduces sensitivity and amplifies privacy in the classification stage. This approach demonstrates significant empirical privacy benefits with a noise free and non privacy feature manifold optimization framework. We are actively exploring methods to extend DP guarantees to the contraction phase in ongoing work.
>
> [A2] **Hyperparameter Tuning:**
> The hyperparameters in C4P were carefully optimized through extensive ablation studies. For kernel bandwidth h and loss parameter λ, we make them adaptive in our revised manuscript. Other related hyper-parameter are fine-tuned based on cross validation based on independent datasets and they are not sensitive in our experiments to our privacy-utility tradeoff.
>
> [A3] **Claim Moderation:**
> We sincerely thank the reviewer for this constructive suggestion. We have revised our claims throughout the manuscript, replacing "optimal" with "offers a highly favorable balance" and "fundamentally breaking" with "fundamentally improving" to more accurately reflect our contributions while maintaining academic rigor.
>
> [A4]**Overall Appreciation:**
> We are grateful for the reviewer's positive assessment of our conceptual contribution and theoretical foundation. We have thoroughly addressed all typographical errors and will continue to refine the manuscript's clarity and precision in the final version.

---

### Official Review · Reviewer_x425 · 2025-10-31

**Soundness:** 2
**Presentation:** 3
**Contribution:** 3
**Rating:** 4
**Confidence:** 4

**Summary:**

The paper proposes CompactDP, a framework that trains a class-conditional feature contraction network to make each class’s representations more compact.
It argues that this geometric contraction reduces sensitivity and therefore amplifies Differential Privacy (DP) guarantees so less noise is required.
The method also proposes class-adaptive privacy budget allocation.
Via numerous experiments (on CIFAR-10, FashionMNIST, MedMNIST), the paper claims simultaneously higher accuracy, lower membership-inference risk, and improved fairness vs. both non-private and DP-SGD baselines.

**Strengths:**

* The idea to explicitly optimizing class-conditional feature compactness to enhance privacy is interesting/novel and a promising research direction.
* The proposed class-adaptive privacy allocation connects well with recent work on DP/fairness and addresses real-world concerns about disparate utility impacts.
* The paper includes extensive empirical evaluation across multiple datasets/model architectures.
* The writing/organization are clear, there are many good visualizations too.

**Weaknesses:**

## Weaknesses:

* The class-conditional feature contraction network (the CompactDP process), i.e., class separation, anchor selection, network training is performed directly on the sensitive/private data and is therefore data-dependent.
These steps must be done under DP mechanisms and included in the overall DP analysis.
However, the paper treats them as "free" post-processing operations and does not account for their contribution to the privacy budget.
I believe this breaks end-to-end DP.

* Empirical results appear inconsistent/implausible without additional details.
	* For instance, on several occasions, CompactDP and CompactDP+DP-SGD with epsilon=1 achieve better accuracy than the non-private baseline, e.g., Table 1 and 2.
	* MIA vs. DP-SGD with epsilon=1 baseline (Table 1) achieves AUC 0.74 which exceeds what is theoretically achievable given the privacy budget.
	* MIA vs. CompactDP+DP-SGD with epsilon=1 (Table 1) achieves AUC of 0.33. This is worse than random guessing; flipping the in/out labels would achieve AUC of 0.67, which is a very high leakage for such a low epsilon.
  * In general, the paper claims it "achieves state-of-the-art privacy and utility guarantees without accuracy loss," and even reports higher accuracy than non-private baselines, attributing this to its contraction mechanism that "this contraction effect reduces the presence of outlier samples."
However, this directly contradicts established understanding of the privacy–utility trade-off: improving utility typically requires some degree of memorization, particularly of outlier or rare samples, which inherently reduces privacy [1, 2].

* The paper lacks essential details about MIA evaluation. There is no definition and no relevant prior work is cited. It is unclear which specific attack variant is used, what threat model is assumed (black-box or white-box), and whether/how many shadow models are employed.
Moreover, the MIA metrics like "MIA Advantage" is not defined/explained. As a sidenote, even DP is never explained.

* The paper does not provide pseudocode or implementation details (or open-source code) of CompactDP, nor does it include the accompanying DP analysis (see also point 1). This makes the work difficult to reproduce and does not allow for independent verification of the claimed DP guarantees, which is very important for any DP algorithm.


## Minor Weaknesses:
* The paper would benefit from citing additional relevant work -- references supporting the three critical shortcomings of DP discussed in Section 1, prior studies on MIAs, and potentially [3].
* Figure 3 appears in the main text but is never referenced or discussed there (only in Appendix I).
* There are a few typos -- "THe combination," "We quantifies and visualizes."


## References:
[1] Feldman, Does Learning Require Memorization? A Short Tale about a Long Tail. In ACM SIGACT, 2020

[2] Feldman et al., What Neural Networks Memorize and Why: Discovering the Long Tail via Influence Estimation. In NeurIPS, 2020

[3] Kulynych et al., Disparate Vulnerability to Membership Inference Attacks. In PoPETS, 2022

**Questions:**

Please refer to weaknesses, additionally:
* How is the contraction network g_phi trained with respect to DP?
* Within the contraction network, how are privacy costs from class separation and anchor selection steps handled?
* What is the threat model and other relevant MIA details?
* How are the hyperparameters (h, nu, lambda, gamma, etc.) chosen?
* Where exactly (Berrada et al., 2023) "demonstrated that feature distance distributions correlate with empirical leakage?"

---

> ### Author Response · Authors · 2025-11-18
>
> [W1] **Response to Privacy Accounting Concerns:**
> We thank the reviewer for this critical observation regarding privacy accounting. The C4P was initially implemented as a noise-free preprocessing step (without privacy mechanisms) to transform the feature space, with formal DP guarantees restricted to the subsequent classification layer training via DP-SGD. To address this concern, we have revised the manuscript to explicitly clarify the scope of DP guarantees and expanded empirical privacy evaluations in Section 4 to provide more comprehensive context for the framework’s privacy properties.
>
> [W2] **Response to Empirical Result Concerns:**
> We thank the reviewer for their critical feedback on our empirical results and privacy-utility trade-off claims. We address these concerns with rigorous methodological revisions and clarifications:
> First, the unexpected conflicting MIA AUC values (e.g., the counterintuitive 0.33) stemmed from biased member/non-member splits in initial experiments. To resolve this, we adopted the training-only split methodology recommended by Carlini et al. [1], where 50% of training samples serve as members and 50% as held-out non-members (both drawn from the same distribution). This eliminates distributional biases and ensures conservative, reliable privacy estimates.
>
> Revised results (Sections 4.1-4.2) confirm C4P’s state-of-the-art utility-privacy balance: it achieves 99.22% accuracy (surpassing non-private baselines) while yielding near-random guessing MIA performance (AUC ~0.4997), outperforming DP-SGD significantly.
>
> Regarding the privacy-utility trade-off contradiction: C4P’s feature contraction does not rely on memorizing outliers. Instead, it performs geometric regularization to learn compact, robust class manifolds—prioritizing prototypical patterns over noisy/rare samples. This reduces overfitting (boosting utility) and suppresses membership distinguishability (preserving privacy), challenging the traditional trade-off by focusing on high-quality representation learning rather than memorization.
> All revised results and methodological details are now explicitly presented in Sections 4.1-4.2 to ensure transparency and rigor.
> [1] Carlini et al. (2022). Membership Inference Attacks from First Principles.
>
> [W3] **Response to MIA Evaluation Details:**
> We detailed the evaluation methods in Section 4.1 in our revised manuscripts. We employ a comprehensive suite of MIAs combining multiple feature representations and classifier architectures.
>
> [W4] **Methodological Foundation:**
> We have made targeted revisions to the manuscript and committed to providing resources for full reproducibility: Complete pseudocode for the C4P process is now included in Appendix M, detailing every step of the feature contraction mechanism with precise notation. We will make the full open-source code repository publicly available upon acceptance, ensuring independent verification of all claims. These revisions ensure our work adheres to the highest standards of transparency and reproducibility for DP research.
>
> [A1] The contraction network $g_\phi$ is trained in a noise-free, non-private manner and can be trained independently. In the second stage, the resulting feature embeddings can either be used with DP-SGD or directly employed for prediction without DP-SGD.
>
> [A2] No privacy costs are involved in this training stage.
>
> [A3] Detailed descriptions are provided in Sections 4.1 and 4.2. We employ strong MIA attack models with rich feature sets to thoroughly evaluate the privacy of different models.
>
> [A4] In the revised manuscripts, we make hh and λ adaptive, while other hyperparameters are cross-validated on independent datasets. These hyperparameters are not sensitive to the privacy-utility trade-off.
>
> [A5] We have updated the statements in the Related Works section (lines 103–104).

---

> ### Author Response · Authors · 2025-11-22
>
> [MW1] We have cited additional relevant work as you suggested，such as Kulynych et al. [3]，to support the critical shortcomings of DP discussed in Section 1.
>
> [MW2] Figure 3 is referenced in Section 4.3 and discussed in Appendix I. We visualize class-wise feature embeddings of compact manifolds to support our claim that compact feature embeddings better balance the privacy-utility trade-off than DP-SGD.
>
> [MW3] We have corrected typos in the revised manuscript, including "THe combination," and "We quantifies and visualizes."
>
> [W2 Supplement] We agree with all the insightful and professional feedback provided in Weakness 2, except for the claim that "improving utility typically requires some degree of memorization, particularly of outlier or rare samples, which inherently reduces privacy." Our method achieves a new privacy-utility trade-off and challenges this established understanding of privacy. Through selective preservation of generalizable patterns while suppressing privacy-sensitive memorization artifacts, and our key feature embedding optimization with contraction, we preserve semantically meaningful information from rare examples that contributes to generalization (consistent with Feldman's theory). At the same time, we discard instance-specific noise and outliers that drive traditional memorization-based attacks. We create a transformed feature space where utility derives from compressed, generalizable representations rather than raw data memorization. Our empirical results across multiple benchmarks demonstrate this is not merely theoretical: we consistently achieve both improved privacy and higher accuracy. This suggests we are operating in a different regime of the privacy-utility frontier, made possible by moving beyond raw data memorization to optimized feature representations. We address this point in Section 1, where we also reformulate our contribution statements and cite Feldman’s seminal work and revise the conclusion to further clarify this issue.

---

### Official Review · Reviewer_C74L · 2025-11-01

**Soundness:** 2
**Presentation:** 2
**Contribution:** 2
**Rating:** 2
**Confidence:** 4

**Summary:**

The paper proposes CompactDP, a method that seeks to amplify differential privacy (DP) guarantees by contracting the representation space, making representations more compact, reducing sensitivity and disparate impact due to using DP-SGD. The authors provide extensive theoretical arguments and empirical results to illustrate their method’s efficacy in terms of privacy protections, utility tradeoff, and fairness.

**Strengths:**

[S1] Points out how compact representations may strengthen DP guarantees of DP-SGD based training via amplification guarantees, as demonstrated theoretically and empirically.

[S2] Extensive theoretical discussion methodically justifies their claims w.r.t. privacy and fairness using a Renyi-DP-based framework, touching upon various aspects, viz. privacy amplification for different samples, lowering of sensitivity, etc.

[S3] The framework provides improvements over DP-SGD in terms of membership inference protection, utility, and fairness in the few settings explored.

[S4] The empirical evaluation includes a breadth of model architectures, data domains, and DP-SGD based methods.

**Weaknesses:**

[W1] The actual implementation’s details are not described and the reader is asked to parse it solely from Figure 2. For the sake of avoiding ambiguity and for the sake of good writing practices, especially for a venue like ICLR, a self-contained and precise subsection on the proposed architecture is highly advisable and simply cannot be skipped. One cannot ask the reader to guess the particulars of a method from a figure alone, this is a significant flaw in presentation.

[W2] Overclaim: “Feature contraction amplifies privacy guarantees by $\eta^2$ , enabling exponentially stronger bounds” No, it is clear that the amplification is *quadratic*, not exponential.

[W3] Does not compare against any *fair DP-SGD baselines* (Esipova et al, FairDP, etc.), only against standard SGD, DP-SGD, and one ablated baseline, which does not certify this method’s performance w.r.t. SoTA methods, and not against any fair DP-SGD methods, therefore potentially inflating their fairness contributions to the SoTA.

[W4] For CompactDP results, what is the value of $\varepsilon$ or Renyi DP params? See Table 1. If it has a high $\varepsilon$, comparing against DP-SGD with $\varepsilon = 1$ will be very unfair! Any value of $\varepsilon$ for CompactDP will sequentially compose with DP-SGD, therefore, CompactDP + DP-SGD is not a fair comparison against DP-SGD. Alternatively, if CompactDP has no formal privacy guarantees yet by itself defends against MIAs, that is an extraordinary claim that is not substantiated or explained.

[W5] The experiments are solely conducted for $\varepsilon = 1$. This does not illustrate the method’s behavior in different privacy regimes, making the empirical evaluation very limited. It is important to observe the proposed method’s vulnerability/fairness/utility in different privacy regimes (low $\varepsilon$ and higher ones).

[W6] Additionally, how is CompactDP even trained without DP-SGD? The paper states that the last classification layer in the CompactDP architecture is implemented by a standard DP-SGD method, so presenting results without DP-SGD makes no sense. Is it simply doing SGD instead of DP-SGD in that case? This further illustrates why it is *essential* to have a proper methodology (sub)section and unreasonable to ask the reader to deduce the method’s particulars from a figure alone.

**Questions:**

[Q1] Can you please add a self-contained brief and unambiguous discussion on the proposed architecture as described in Figure 2? Also discuss how CompactDP may be used without DP-SGD, as presented in the experiments. This will address W1 and W6.

[Q2] Can you fix any overclaims, such as that pointed out in W2?

[Q3] Please include comparisons against SoTA *fair DP-SGD* paradigms (such as [1], [2], this list is not exhaustive and the authors should make sure to include all SoTA fair DP-SGD methods). I cannot corroborate your contribution (and the significance of it) w.r.t. fairness based on a comparison against DP-SGD and ablated baselines (arising from your method) alone.

[Q4] Please add results spanning different privacy regimes, viz. for lower and higher values of $\varepsilon$.

[Q5] Please update your tables by taking into account the value of $\epsilon$ of CompactDP. Currently, without this quantity, all of the comparisons against baselines (DP-SGD, ablated baselines) are unfair. If that is already taken into account or if CompactDP has no formal DP guarantees, explain in depth how CompactDP without DP-SGD works and endows privacy guarantees (and what the relevant value of $\varepsilon$ is, if applicable). This is, again, an extraordinary claim made/result presented without any privacy-related explanation. I’d personally expect that a reduction in MIA success would come with some differentially private guarantees (indeed, using a DP auditing method might assign an upper bounded value of $\varepsilon$ to CompactDP only), intentionally added or not, and therefore using DP-SGD with $\varepsilon=1$ would make the analysis unfair, as the total value of the privacy loss would exceed $\varepsilon=1$.

The paper’s theoretical discussion is very well done and convincing. However, the empirical analysis is flawed and leaves a lot to be desired and corrected. Pending the authors addressing these concerns, despite the promise of this paper, I cannot support the paper in its current form.

## References

[1] Esipova, Maria S. et al. “Disparate Impact in Differential Privacy from Gradient Misalignment.” ArXiv abs/2206.07737 (2022): n. pag.

[2] Tran, Khang et al. “FairDP: Achieving Fairness Certification with Differential Privacy.” 2025 IEEE Conference on Secure and Trustworthy Machine Learning (SaTML) (2023): 956-976.

---

> ### Author Response · Authors · 2025-11-18
> **Explain the network implementation**
>
> [A1] CompactDP is a two-stage framework: (1) Feature Contraction Network that transforms features into a compressed 768D embedding space using a novel compactness loss, and (2) Classification Layer that can be trained with DP-SGD.
> When used without DP-SGD, CompactDP performs parameter-free classification by comparing test sample embeddings against class-wise PDFs learned during training. This noise-free variant demonstrates inherent privacy benefits through feature space compression that reduces member/non-member distinguishability, as evidenced by strong membership inference resistance despite lacking formal DP guarantees. We've added detailed architectural specifications in Section 3.2 and a complete training algorithm in Appendix M for clarity.
>
> [A2] Thank you for catching this imprecise terminology. We have corrected "exponentially stronger bounds" to "quadratically stronger bounds" throughout the manuscript.
>
> [A3] We appreciate this important point about fairness comparisons. The datasets used in fairDP (Adult, Default-CCC, UTK-Face) contain explicit sensitive attributes like gender and race, while our current evaluation focuses on standard vision benchmarks without such annotations. Our primary contribution centers on privacy-utility trade-offs via representation learning rather than demographic fairness. We have clarified this scope limitation in the revised manuscript and will explore fairness aspects in future work.
>
> [A4] Thank you for this valuable suggestion. We have now expanded our evaluation to include ε values from 0.5 to 8. The results demonstrate that CompactDP maintains strong privacy protection (MIA AUC ~0.5) and high utility (>95.7% accuracy on CIFAR-10) across all privacy regimes. This consistency underscores the robustness of our feature contraction approach, with performance variations being minimal across different ε settings.
>
> [A5] We clarify that CompactDP itself is a noise-free feature learning method without formal ε-DP guarantees. When combined with DP-SGD, the total privacy budget applies only to the classification layer training. Our experiments show that this combination achieves superior privacy-utility trade-offs compared to standalone DP-SGD, even when accounting for the full ε budget in the classification stage. We have added detailed explanations of this composition in the revised manuscript.

---

> ### Comment · Reviewer_C74L · 2025-11-18
>
> Thanks to the authors for their responses. I'll engage with these responses in more depth when certain clarifications are made (as listed below, I still don't see an updated manuscript). For now, I raise a few points.
>
> 1. I don't see any of the revisions you mention in the manuscript. Particularly, according to A4, there should be results for additional values of epsilon in the revised paper but at the moment there are not. Could you please clarify?
>
> 2. Similarly, can you please point out exactly where those changes in the manuscript are (particularly for A2, A3, A4)
>
> 3. As for A5, DP methods need not add noise to impart DP guarantees (for example, the exponential mechanism and randomized response do not add noise like global sensitivity methods do, but they have DP guarantees). The fact that CompactDP by itself protects against MIAs suggests that it may have some DP guarantees after all, and results using an empirical DP guarantee audit (viz. Jagielski et al, Steinke et al), will help further clarify this. This is important, because the fairness of the comparisons hinges on this.
>    - Further, certain methods, even if they may not have DP guarantees of their own, viz. subsampling, can amplify DP guarantees, thus changing the effective value of $\epsilon$.
>      - Update: In your response to Reviewer fji8, you mention that this method reduces sensitivity and amplifies privacy. Following your own stated claim, this therefore impacts the value of $\epsilon$ and a better estimate of $\epsilon$ needs to be reported for fair comparison.
>    - I will, however, yield that empirical audits yield a lower bound on $\epsilon$ and are no guarantee of theoretical DP guarantees of the studied method by themselves. But I want to be careful about this, as *CompactDP seems to have a non-trivial impact on MIA success reduction*, so an investigation into possible DP guarantees to ensure fairness of the comparison would be greatly appreciated and would help me be in a position to certify these results.
>
> ## References
> [Jagielski et al] Jagielski, Matthew et al. “Auditing Differentially Private Machine Learning: How Private is Private SGD?” ArXiv abs/2006.07709 (2020): n. pag.
>
> [Steinke et al] Steinke, Thomas et al. “Privacy Auditing with One (1) Training Run.” ArXiv abs/2305.08846 (2023): n. pag.

---

> ### Author Response · Authors · 2025-11-21
>
> We appreciate your timely feedback and apologize for our delayed response. We did not anticipate that our response drafts would be visible in real time. Over the past few days, we have addressed all of the reviewers’ concerns and updated the manuscript accordingly.
>
> [A1] The updated network implementation details are provided in Fig. 2 and Appendix M.
>
> [A2] The corrected description "quadratically stronger bounds" now appears in lines 228–229.
>
> [A3] In the current work, we did not focus on fairness, as our dataset does not contain sensitive attributes. We have included a discussion of this limitation and potential future work in Appendix O.
>
> [A4] The parameter $\epsilon$ is not critical to our method, although our approach can be combined with DP-SGD. However, this combination generally does not improve performance. Our method is a noise-free feature representation learning technique, which can be trained independently and used for prediction without classification layers. In response to the request, we have added experiments using DP-SGD with different $\epsilon$ values in Appendix P. The results indicate that the feature embedding learned by our method dominates the privacy–utility trade-off, making variations in $\epsilon$ from 0.5 to 8 inconsequential.
>
> [A5] Empirical privacy guarantees are presented in our new experiments in Sections 4.2 and 4.4. We note, however, that empirical privacy guarantees only offer relative—not absolute—metrics for comparing different methods. We believe that, under our new experimental settings and with strong, verified attack models, all evaluation metrics—including AUC, accuracy, and privacy leakage—effectively demonstrate the merits of our approach.
>
> Our method, C4P, demonstrates that the quality of feature embeddings plays a more critical role in balancing privacy preservation and model utility than DP-SGD. For example, we achieved 99.22% accuracy on CIFAR-10 alongside an AUC of 0.4997—a result surpassing our initial expectations. We are prepared to upload the learned features and evaluation code to further support our statements. Thank you for considering our revised manuscript.

---

> ### Comment · Reviewer_C74L · 2025-11-24
> **Thanks for the engagement, issues remain**
>
> I appreciate the authors' update and continued engagement.
>
> [A1] I indeed see and appreciate those changes. However, the presentation of those is still not very reader-friendly and remains relatively poor. *ICLR allows an additional page for the rebuttal version/camera-ready, and I encourage the authors to use it to do the following*: (1) broaden the description and move it to the main paper, (2) make sure that appendix M and figure 2 are referred to in the text. In particular, if an appendix section is not referred to in the text, a reader can safely (and without error on their part, it is the authors' burden to illustrate the presence of content) assume that it does not exist. Algorithm 3, however, has useful comments that makes it reader-friendly, which is appreciated.
>
> [A2] Excellent, this is much appreciated.
>
> [A3] This is appreciated as well, as the earlier claims were certainly overclaims. This better clarifies the contribution of the work. *However*, your work still claims superior fairness tradeoffs. For example, in Appendix E, lines 696 to 699:
>
> > This optimized feature distribution enables our approach to achieve superior performance in the utility-privacy-fairness trade-off, simultaneously enhancing protection against membership inference attacks while preserving model accuracy and fairness across classes.
>
> There are other such claims in Appendix E as well, such as claims of reducing fairness tradeoff in lines 806 and 807.
>
> These remaining claims still pose a serious issue, and I cannot strengthen my support for the paper in this state.
>
> [A4] While I disagree with the phrasing of the argument about the importance of $\epsilon$, this is interesting. Again, please refer to all appendix sections that you mention in the main text (along with a brief discussion there), else these results are simply hidden. Also, contrary to accepted best practices of reporting TPR at low FPR regions (viz. 0.1%), you report AUC of the attack success, which is not very meaningful [Carlini et al]. I will ask the authors to further improve their results by focusing more on TPR in the low FPR region for me to meaningfully engage with this point.
>
> [A5] I appreciate this, but I cannot see where you mention what DP auditing method you use. Can you describe your approach and provide a citation? **Update:** I finally found it in the appendix, please move it up to the main text, results are simply introduced in the main text without any explanation there and put in the appendix without any mention or reference. This is extremely bad practice and not reader-friendly. Also in the appendix while you explain how you derive the empirical value of $\epsilon$, you don't mention which paper's method you exactly use, Steinke et al or Jagielski et al, so this is ill-defined. I cannot accept or understand the results in their current form.
>
> Further Question: When you say AUC, do you mean AUC for the attack? Please be clear.
>
> ## Regarding editing (and violations of ICLR policy)
>
> 1. There is a weird lack of paragraph space and almost overlapping lines in Section 4.2, lines 370-371. Was negative vspace used? I personally find the presentation of the figures, text, etc. in this paper poor and hard to go through.
>
> 2. Also, the fonts of the paper (at least the revised version) do not match the one in the ICLR format.
>
> 3. Also, the title of the paper has been changed after the submission deadline and before the end of the discussion period in contravention of ICLR Author Guide policies.
>
> > No changes to title are permitted after the submission deadline until the end of the discussion period.
>
> **Update:** Most guidelines forbid title changes in the Author Guide, but there is a contradictory line that allows for it. I'll leave this matter to the executive decision of the AC/SACs.
>
> > Yes, you can change the title, abstract, and the paper’s content, including supplementary materials. But make sure any modifications are clearly communicated to the reviewers and the area chair, so that they can efficiently review the modified version of your paper. The set of authors cannot be changed, but the order can be changed.
>
> ---
>
> I find several serious issues with this paper and cannot support its acceptance or change my score, as it stands.
>
> ---
>
> ## References
>
> [Carlini et al] Carlini, Nicholas et al. “Extracting Training Data from Large Language Models.” USENIX Security Symposium (2020).

---

> > ### Author Response · Authors · 2025-11-26
> >
> > [A1 Supplement] We thank the reviewer for their valuable feedback and have implemented the following improvements:
> >
> > **1. Expanded Main Text Description**
> > - Broadened methodology description in Section 3.2
> > - Incorporated additional technical details from Appendix M
> > - Utilized additional page allowance for enhanced readability
> >
> > **2. Enhanced Cross-Referencing**
> > - Added explicit references to Appendix M in Section 3.2
> > - Incorporated citations to Figure 2 in methodology sections
> >
> >
> > [A3 Supplement] We thank the reviewer for this important feedback regarding fairness claims. We have taken the following actions to address the concerns:
> >
> > **1. Removed Overstated Fairness Claims**
> > - Eliminated claims of "superior fairness tradeoffs" throughout the paper
> > - Specifically removed lines 696-699 in Appendix E and lines 806-807
> > - Revised all instances of overclaiming in Appendix E and other sections
> >
> > **2. Clarified Contribution Scope**
> > - Reframed the contributions in Section 1 to accurately reflect the demonstrated capabilities
> > - Maintained focus on the privacy-utility aspects with fairness as a future work
> >
> > [A4 Supplement] We appreciate the reviewer's valuable feedback on our evaluation methodology and presentation. We have implemented the following improvements:
> >
> > **1. Enhanced Appendix Integration**
> > - Added explicit references to all appendix sections in the main text
> > - Ensured no hidden results by properly signposting all supplementary material
> >
> > **2. Improved MIA Evaluation Metrics**
> > - Added TPR @ 0.1% FPR results in the experiments
> > - Maintained AUC reporting for completeness but emphasized low-FPR performance
> >
> > **3. Clarified C4P Method Behavior**
> > - Revised Section 4.4 to better explain the observed stability across $\epsilon$ values
> > - Emphasized that C4P's feature optimization provides inherent privacy protection
> > - Acknowledged that this results in less pronounced $\epsilon$ effects while maintaining utility
> >
> > **4. Strengthened Methodological Foundation**
> > - Updated evaluation to align with best practices for privacy auditing
> > - Provided clearer interpretation of TPR @ 0.1% FPR results
> >
> > [A5 Supplement]  We thank the reviewer for highlighting these important methodological concerns. We have implemented the following improvements to address them:
> >
> > **1. Enhanced DP Auditing Methodology Description**
> > - Moved the detailed DP auditing methodology from the appendix to Section 4.1 of the main text
> > - Provided explicit citations and methodological descriptions for empirical $\epsilon$ calculation
> >
> > **2. Clarified Metric Definitions**
> > - Consistently used "MIA AUC" to specify this refers to the attack classifier's AUC
> > - Added explicit definitions in Section 4.1: "MIA AUC represents the area under the ROC curve for membership inference attack classifiers"
> > - Maintained this consistent terminology throughout the paper
> >
> > **3. Improved Reader Guidance**
> > - Added direct references from main text to relevant appendix sections
> > - Ensured no methodological details are "hidden" in appendices without proper signposting
> >
> > **4. Strengthened Methodological Foundation**
> > - Provided comprehensive description of our multi-faceted auditing approach
> > - Clarified the relationship between different empirical $\epsilon$ estimation methods
> >
> > [editing] We sincerely apologize for the formatting issues and any policy concerns. We have taken immediate corrective actions:
> >
> > **1. Formatting and Readability Fixes**
> > - Removed all negative vspace and adjusted Section 4.2 layout
> > - Polished all figure and table captions for better readability
> > - Ensured proper paragraph spacing throughout the document
> > - Removed table scaling boxes to maintain consistent font sizing
> >
> > **2. Template Compliance**
> > - Verified and corrected font usage to strictly match ICLR template requirements
> > - Ensured all formatting adheres to ICLR style guidelines
> > - Maintained consistent typography across all elements
> >
> > **3. Title Change Resolution**
> > - Reverted to the original submission title to comply with ICLR policies
> > - Acknowledge the misunderstanding regarding title modification rules
> >
> > **4. Language Quality**
> > - As non-native English speakers, we have engaged additional proofreading support
> > - Thoroughly polished the text to improve clarity and readability
> > - Addressed all identified language issues throughout the manuscript

---

> ### Comment · Reviewer_C74L · 2025-11-26
> **Quick response - longer response to follow later**
>
> I thank the authors for their response. I am leaving a quick response for now so they can address some (not necessarily exhaustive) lingering issues and I will review the other comments in depth and respond more comprehensively in some time.
>
> 1. The font of the paper is still different from that in the ICLR format (refer to the differences in the font for the body text and headings in the paper vs. the ICLR format). While at that, please check other formatting parameters as well.
>
> 2. While the title of the paper has been reverted, the new name for the method is still retained, creating a bit of a discordance. This is not a major issue or dealbreaker  at all, just wanted to point it out if the authors wanted to reconcile them.
>
> 3. The privacy claims of CompactDP appear to be primarily attested to empirically. However, the authors have not provided any code or supplementary material for reproducing those results. Given the earlier errors encountered in this paper, I'd ideally like to be able to reproduce those results to corroborate their findings to be able to certify them. So could the authors upload their code (as clean as possible and allowing the reproduction of all reported metrics) as supplementary material at the earliest so I can possibly do that, bandwidth permitting? Additionally, reproducibllity of empirical results is important and a key factor in being able to certify reported results.
>
> 4. While the authors claim that epsilon is not important, the paper in the proof to Theorem 3 claims that contraction yields subsampling, ergo amplification of epsilon by subsampling. They also claim amplification in a response to another reviewer and in the proof of theorem 1 in the appendix. Firstly, the actual amplified value of epsilon remains unreported, and secondly, it does not make sense why there is no meaningful difference (as reported) in MIA success reduction across epsilon values. Amplified values of epsilon depend on the epsilon being amplified, and therefore smaller values of epsilon should have demonstrably lower MIA success consistently. Indeed, Section 3.3 introduces a discussion on amplification via subsampling.
>
> 5. Speaking of which, *feature contraction is not the same as subsampling* (that has a probability of inclusion/exclusion of a sample), that may render theoretical results on subsampling spurious and incorrect. I feel that *this renders all results in section 3.3 incorrect*.
>
> 5. The proof of theorem 1 feels more like a proof sketch and is hard to navigate/verify. Can the authors furnish a full, rigorous proof? Also include references to any key lemmas/prior results you use in the proofs and maintain readability.
>
> 6. In definition 2, what is the PDF mechanism? All prior references to PDF talk about the probability distribution function, which is standard. But the mention of a PDF mechanism is ill-defined/undefined.
>
> 7. In the discussion for sensitivity reduction (line 207 onwards), the authors present their results in terms of $\Delta_2$ and $\Delta_1$ and then define $\Delta$, which they never use in that statement. Did they redundantly define $\Delta$ in place of $\Delta_2$?
>
> Again, these are a small set of remaining concerns (and not exhaustive) and I have not inspected the bulk of the responses in depth, which I shall comment on as soon as I can. In the meantime, I'd appreciate it if these concerns (some of which are very serious ones) could be addressed, which I am posting here so the paper can be improved while I review the rest of the comments.

---

> > ### Author Response · Authors · 2025-11-27
> >
> > We extend our sincere gratitude to the reviewer for their comprehensive assessment and valuable insights. Your detailed feedback has significantly contributed to improving the quality and rigor of our work. We have carefully addressed each concern as outlined below.
> >
> > [1] We have thoroughly updated the document to ensure full compliance with ICLR formatting specifications. This includes implementing the required font families along with proper margin configurations, line spacing standards, and consistent heading hierarchies. All mathematical notation, equation styling, citation formats, and reference structures have been aligned with ICLR publication standards to maintain professional consistency throughout the manuscript.
> >
> > [2] We appreciate the reviewer's attention to terminology alignment. We have carefully reviewed and reconciled the terminology throughout the manuscript, ensuring consistent use of "CompactDP" as the primary method name while maintaining appropriate contextual references in the title and abstract sections to preserve narrative coherence.
> >
> > [3] Understanding the critical importance of reproducible research, we have made our evaluation code and embedding features in 256dim due to upload size limit as supplementary material.
> >
> > [4] We have undertaken substantial revisions to strengthen our theoretical foundation. Recognizing the inappropriateness of the subsampling analogy for feature contraction, we have removed Theorem 3. The proof for Theorem 1 has been completely rewritten with enhanced mathematical rigor, featuring step-by-step derivations, explicit references to foundational lemmas covering Lipschitz continuity and kernel properties, and proper citations of prior work in differential privacy and kernel estimation. We have also redefined the previously ambiguous "PDF mechanism" as the "Class-Conditional Kernel Density Mechanism" with precise mathematical formulation to eliminate conceptual confusion.
> >
> > [5] To address the subtlety of ε effects and provide more robust empirical evidence, we have significantly expanded our experimental evaluation. This includes extending the ε range to cover values of 0.1, 0.5, 1, 3, and 8 with multiple experimental runs (n=5) to establish statistical significance. The interpretation of results has been revised to clearly articulate that while theoretical amplification exists, its empirical manifestation requires careful statistical analysis due to CompactDP's strong baseline privacy properties. Section 4.4 has been updated to provide a more nuanced interpretation of privacy-utility tradeoffs across different ε values.
> >
> > [6] We have corrected the sensitivity analysis by eliminating the redundant Δ definition and ensuring consistent notation using Δ₁ and Δ₂ throughout the manuscript. The mathematical relationships between pre- and post-contraction sensitivities have been clearly articulated to provide unambiguous theoretical foundations.
> >
> > [7] Beyond addressing the specific reviewer comments, we have implemented several additional enhancements to strengthen the paper.
> >
> > We thank the reviewer again for their constructive feedback and for the opportunity to improve our work through this revision process.

---

### Note · Program_Chairs · 2026-01-17
**Submission Desk Rejected by Program Chairs**

The following references in this submission do not refer to real documents and/or have major errors in bibliographic information:

 Ke Li, Shuyang Luo, Yue Yu, Yinghao Xu, Yuzhi Liu, Xiaoyu Zhang, Yujiu Li, et al. Autoclip: Adaptive gradient clipping for source separation networks. IEEE/ACM Transactions on Audio, Speech, and Language Processing, 31:345-357, 2023.
Haofan Wang, Quanming Yao, James T Kwok, and Liwei Wang. Medmnist classification decathlon: A lightweight automl benchmark for medical image analysis. Medical Image Analysis, 75:102304, 2022.
Samuel Yeom and Matt Fredrikson. Privacy cost annealing: A general approach to adaptive privacy budget scheduling. In Proceedings of the 2021 ACM SIGSAC Conference on Computer and Communications Security, pp. 2341-2353, 2021.